# ViPER: Empowering the Self-Evolution of Visual Perception Abilities in Vision-Language Models

**Juntian Zhang**[1,2], **Song Jin**[1,2], **Chuanqi Cheng**[1], **Yuhan Liu**[3], **Yankai Lin**[1], **Xun Zhang**[2],
**Yufei Zhang**[2], **Fei Jiang**[2], **Guojun Yin**[2], **Wei Lin**[2]*, **Rui Yan**[4]*
[1]Gaoling School of Artificial Intelligence, Renmin University of China, [2]Meituan,
[3]MBZUAI, [4]Wuhan University
{zhangjuntian, jinsong8, chengchuanqi, yuhan.liu}@ruc.edu.cn

## Abstract

The limited capacity for fine-grained visual perception presents a critical bottleneck for Vision-Language Models (VLMs) in real-world applications. Addressing this is challenging due to the scarcity of high-quality data and the limitations of existing methods: supervised fine-tuning (SFT) often compromises general capabilities, while reinforcement fine-tuning (RFT) prioritizes textual reasoning over visual perception. To bridge this gap, we propose a novel two-stage task that structures visual perception learning as a coarse-to-fine progressive process. Based on this task formulation, we develop **ViPER**, a self-bootstrapping framework specifically designed to enable iterative evolution through self-critiquing and self-prediction. By synergistically integrating image-level and instance-level reconstruction with a two-stage reinforcement learning strategy, ViPER establishes a closed-loop training paradigm, where internally synthesized data directly fuel the enhancement of perceptual ability. Applied to the Qwen2.5-VL family, ViPER produces the **Qwen-Viper** series. With an average gain of **1.7%** on seven comprehensive benchmarks spanning various tasks and up to **6.0%** on fine-grained perception, Qwen-Viper consistently demonstrates superior performance across different vision-language scenarios while maintaining generalizability. Beyond enabling self-improvement in perceptual capabilities, ViPER provides concrete evidence for the reciprocal relationship between generation and understanding, a breakthrough to developing more autonomous and capable VLMs.

## 1 Introduction

Multimodal Large Language Models (MLLMs) have exhibited strong capabilities and wide-ranging application value across diverse domains (Caffagni et al., 2024). In contrast to Large Language Models (LLMs), MLLMs extend beyond text to multiple modalities, enabling them to handle more complex, information-rich tasks and establishing them as critical for embodied AI and world models (Yin et al., 2024). As an important type of MLLMs, Vision-Language Models (VLMs) are typically architected with a visual encoder to process pixel-level information and a language model backbone for semantic understanding, establishing a connection between the two modalities (Li et al., 2024; Wang et al., 2025b). Consequently, the overall performance of a VLM depends on the coordinated functioning of its visual encoder and language backbone.

Recent progress, such as o1 (El-Kishky, 2024) and Deepseek-R1 (Ren et al., 2025), has advanced the reasoning ability of LLMs, leading to increased focus on "slow thinking" (Wang et al., 2025c). Accordingly, the field has led to the development of multimodal reasoning models (Huang et al., 2025a; Li et al., 2025). Although Chain-of-Thought methods (Wei et al., 2022) have improved the reasoning ability of these models, the expressive limitations of the text present inherent constraints (Lyu et al., 2024; Stechly et al., 2024). For many vision-language tasks, especially those demanding fine-grained perception, performance cannot be attributed solely to the power of the language model

---

* Corresponding author.

backbone. These tasks hinge critically on sophisticated visual understanding coupled with language reasoning, a dual requirement that poses a significant challenge for current VLMs (Shao et al., 2024; Hu et al., 2024). This interdependence means that post-training efforts which enhance only the visual or linguistic component in isolation yield marginal improvements, underscoring the need for approaches that enable their co-evolution.

Several recent studies have proposed task-specific architectures to address the challenges of fine-grained visual perception. However, these approaches often exhibit notable limitations. A widely adopted strategy (Udandarao et al., 2024; Hou et al., 2025; Zhang et al., 2025c; Cao & Ou, 2025)involves scaling up training data by distilling knowledge from superior models. This paradigm is computationally inefficient due to the high cost of data synthesis, and large-scale distillation can lead to marginal performance gains while compromising generalization, resulting in an unsustainable trade-off. In contrast, another line of research employs reinforcement learning (RL) within a "thinking-with-image" paradigm , where performance is improved through iterative tool use and multi-step interaction (Zheng et al., 2025b; Shao et al., 2024; Hu et al., 2024; Gao et al., 2024; Zhou et al., 2024). While this improves system-level outcomes, it introduces substantial latency due to multi-round interactions. Moreover, the reasoning process often becomes superficial, emphasizing tool manipulation over visual comprehension, and fails to improve the model's underlying perceptual capabilities on downstream tasks (Su et al., 2025a; Zhang et al., 2025a; Sun et al., 2024).

To address these challenges, we formulate the enhancement of visual perception as a two-stage structured process. The first stage cultivates holistic image reasoning and static scene understanding via self-critical caption refinement, teaching the model to "see widely". The second stage focus on fine-grained perception and dynamic change awareness by inferring visual operations from image differences, training the model to "focus accurately". This deliberate coarse-to-fine progression structures the model's learning trajectory from global understanding to local precision.

Correspondingly, we introduce **ViPER**, a self-evolutionary framework that concretizes the two-stage task by integrating data synthesis and the RL method. ViPER consists of two components: an automated data synthesis module and a closely integrated two-stage RL method. By incorporating both image-level and instance-level reconstruction, ViPER's data synthesis establishes a bidirectional mapping between visual and textual modalities at dual-granularities, intrinsically using the generation process to solidify perceptual understanding. The self-synthesized data is seamlessly channeled into a tightly coupled two-stage RL process. This integration creates a self-reinforcing loop where generation and learning are intertwined, eliminating the need for external bootstrapping and thereby establishing a self-evolving paradigm.

Based on ViPER, we constructed the *Viper10K* dataset and yield **Qwen-Viper** series perceptually enhanced from Qwen2.5-VL. Experiments revealed that during training, the models spontaneously developed a "thinking-with-image" capability and learned to redirect attention to critical details. On seven comprehensive benchmarks encompassing single-image, multi-image, and hallucination tasks, the Qwen-Viper series demonstrated **consistent gains** over the baseline. Notably, on fine-grained perception tasks, the 3B and 7B models achieved significant improvements of up to **4.4%** and **6.0%**, respectively. These results validate that ViPER enables VLMs to undergo self-evolution on perception-centric tasks.

In summary, our main contributions are threefold:

I. We designed **a progressive two-stage task** that guides VLMs from holistic scene reasoning to fine-grained visual understanding, establishing a structured learning paradigm for perceptual self-improvement.

II. We propose **ViPER**, a self-evolutionary paradigm that bootstraps VLM perception through a closed loop of data construction and RFT, producing *Viper10K* and the Qwen-Viper models. [1].

III. We demonstrate that Qwen-Viper achieves **consistent improvements** on perception-intensive tasks, revealing the reciprocal relationship between generation and understanding by practice.

---

[1]We open-source the implementation codes and data at https://github.com/Icarus1216/ViPER

## 2 RELATED WORK

### 2.1 VISION-LANGUAGE MODEL

The rapid advancement of LLMs has catalyzed substantial progress in VLMs (Yang et al., 2025; Liu et al., 2024a;b) . Representative proprietary models include GPT-4o (Hurst et al., 2024), Gemini 2.5 (Comanici et al., 2025), and Claude3 (Anthropic, 2024). Among open-source models, LLaVA (Liu et al., 2023) has gained widespread adoption through its three-stage architecture, which comprises a Vision Transformer (ViT) (Dosovitskiy et al., 2020), a connector module and an LLM. Subsequent research has largely extended and refined this architecture. For example, LLaVA-OneVision (Li et al., 2024) introduced a dynamic resolution mechanism that adaptively adjusts the number of visual tokens based on the input image resolution. Similarly, Qwen2-VL (Wang et al., 2024a) also adopts this encoding strategy and incorporates the M-RoPE positional encoding mechanism, enabling unified processing of 1D text, 2D images and 3D video data. Its successor, Qwen2.5-VL (Bai et al., 2025) further integrates sparse attention mechanisms within the visual encoder architecture. DeepSeek-VL Lu et al. (2024) optimizes encoding through dual encoders that separate visual and textual information for various downstream tasks. The InternVL series (Zhu et al., 2025; Wang et al., 2025b) proposed using a thumbnail to integrate global image information without substantially increasing the number of tokens.

### 2.2 REINFORCEMENT LEARNING-ENHANCED MULTI-MODAL REASONING

The integration of reinforcement learning (RL) has substantially enhanced the reasoning capabilities of LLMs, as demonstrated by models such as DeepSeek-R1 (Guo et al., 2025) and Kimi-K1.5 (Team et al., 2025). This progress is driven by effective RL frameworks, including DPO (Rafailov et al., 2023), PPO (Schulman et al., 2017), GRPO (Guo et al., 2025), GSPO (Zheng et al., 2025a), and DAPO (Yu et al., 2025a). Recent work has successfully extended these methodologies to VLMs, leading to promising results in systems such as Vision-R1 (Huang et al., 2025a) and VLM-R1 (Shen et al., 2025). Alongside end-to-end RL training paradigm, a significant research direction focuses on tool-augmented methods for multi-modal reasoning (Zhang et al., 2025b; Huang et al., 2025b; Su et al., 2025b). The success of multi-turn tool-calling in advanced agents such as OpenAI's O3 (OpenAI, 2025) has inspired numerous efforts to equip models with specialized visual tools. Representative works include DeepEyes (Zheng et al., 2025b), Chain-of-Focus (Zhang et al., 2025d), and Mini-o3 (Lai et al., 2025) aiming to improve the models' ability to dynamically invoke tools for solving complex reasoning tasks.

In contrast to these methods, our proposed method ViPER integrates data construction with post-training process to form a closed-loop cycle. The framework employs dual-granularity reconstruction processes to promote understanding by generation, enabling self-driven evolution without external supervision or large-scale data.

## 3 METHODOLOGY

In this section, we first formalize the two-stage tasks in Section § 3.1. Then we elaborated on the proposed self-evolutionary framework ViPER in Section § 3.2 and introduced its two key components: the data synthesis module and the coupled two-stage reinforcement learning strategy.

### 3.1 TASK FORMULATION

**Caption Self-Refining**: The first stage involves the self-refinement of captions. This task is designed to train a unified VLM to revise errors or biases in its own generated textual descriptions through deconstructing visual information and self-reflection.

Specifically, for each training sample $(I, C_g, R) \in \mathcal{D}$ , the model performs the task of self-correction. It takes the image $I$ and the initial caption $C_g$ which is generated by itself with parameters $\theta_0$ as input, analyses inaccurcy in the original caption and outputs a set of refinement actions:

$$R_{\text{pred}} = f(I, C_g; \theta). \tag{1}$$

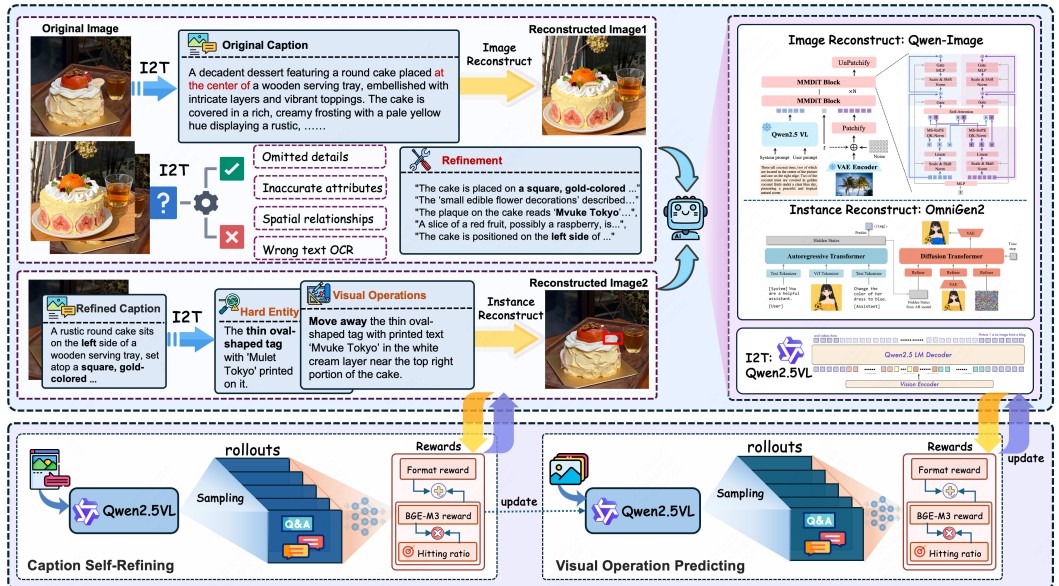

Figure 1: An overview of ViPER. The upper part illustrates the two-stage data synthesis framework, whose core component is a bidirectional vision-language mapping module composed of a VLM and a diffusion model. The lower part depicts the corresponding two-stage RL process: the first stage focuses on caption self-refinement, and the second stage on visual-operation prediction. The Qwen2.5-VL models are trained on *Viper10K* constructed by the ViPER framework and result in Qwen-Viper, thereby achieving substantial self-improvement.

The goal of the training is to minimize the discrepancy between the predicted set of refinement points $R_{\text{pred}}$ and the ground-truth set $R$ by adjusting the parameters $\theta$. This is formulated as minimizing the expectation of a discrepancy metric $\delta(R, R_{\text{pred}})$ over the training distribution:

$$\min_{\theta} \mathbb{E}_{(I, C_g, R) \sim \mathcal{D}} \left[ \delta \left( R, R_{\text{pred}} \right) \right]. \tag{2}$$

Through this process, the VLM learns to iteratively refine its own generated captions through an image-based reasoning process. This task is designed to enhance the model's ability for self-reflection and correction in visual perception, without relying on external correction mechanisms.

**Visual-Operation Predicting**: The second phase aims to train the VLM to predict visual operations based on a pair of highly similar images with slight differences in visual information. The reconstructed image is generated by applying an editing operation to the original image, and the instruction itself is generated by the same VLM with parameters $\theta_1$. For each training sample $(I_{\text{orig}}, I_{\text{recon}}, Ops) \in \mathcal{D}$, the VLM predicts the visual operations based on careful perception and reasoning across the image pair:

$$Ops_{\text{pred}} = f(I_{\text{orig}}, I_{\text{recon}}; \theta), \tag{3}$$

Where $Ops_{\text{pred}} \in \mathcal{T}$ is the model's predicted visual operations. The objective of the training process is to minimize the discrepancy between the model's predicted visual operations and the actual visual operation instructions:

$$\min_{\theta} \mathbb{E}_{(I_{\text{orig}}, I_{\text{recon}}, Ops) \sim \mathcal{D}} \left[ \delta \left( Ops, Ops_{\text{pred}} \right) \right], \tag{4}$$

Where $\delta : \mathcal{T} \times \mathcal{T} \to \mathbb{R}^+$ is a text sequence difference metric. Through this process, the VLM learns to predict accurate visual operation instructions based on the differences between image pairs, thereby enhancing the model's ability to focus on key information and understand visual operations. The two-stage tasks, encompassing both static scene understanding and dynamic visual reasoning, not only enhance the model's image-based planning and reasoning abilities but also strengthen its fine-grained visual perception.

## 3.2 ViPER: The Self-Evolutionary Framework

In this section, we introduced the unified framework **ViPER**, which integrates the data synthesis process with a corresponding two-stage RL strategy, and elaborated on how it enables self-evolutionary enhancement of perception capabilities.

### 3.2.1 The Data Synthesis Module

A central part of ViPER is an automated data synthesis module, which functions as an integrated system for two-stage training data generation. This system seamlessly connects the upstream *Caption Self-Refining* task to the downstream *Visual-Operation Predicting* task. The structure of this module is illustrated in the upper section of Figure 1.

The first-stage data synthesis is centered on an image-level reconstruction process that creates a closed-loop feedback mechanism for the VLM. The VLM first generates a static description from the original image. A diffusion model then reconstructs an image based on this description, which, due to the inherent information loss in visual-to-text conversion, exhibits local discrepancies from the original. These visual differences serve as a form of visual feedback, guiding the VLM to critique and refine its initial description by rectifying errors in object attributes, text, and spatial relationships. Thus, the diffusion model acts not merely as a generator, but as a critic that enables the VLM to iteratively optimize its visual grounding.

In contrast, the second-stage synthesis shifts the focus to instance-level reconstruction. Leveraging the refined caption from the upstream task, the VLM selects hard entities and generates corresponding visual operation instructions through a hand-crafted heuristic rules, as detailed in Appendix A.2. A diffusion model then executes these instructions to edit the original image, producing a reconstructed version. The instructions used naturally serve as ground truth for the *Visual-Operation Predicting* task, where the VLM learns to infer the operation from the image pair. Crucially, by concretizing instructional intent into observable visual variations, the generative model enables the VLM to learn fine-grained visual reasoning from self-induced scene changes.

The introduction of the generative model imbues the process with profound significance: It externalizes the VLM's internal reasoning into a concrete visual snapshot. This act of mapping critical cognitive steps back into the visual modality provides a tangible "cognitive anchor" for the VLM. Effectively, it endows the model with a form of image imagination, allowing it to perceive, critique, and refine its own understanding by confronting the visual consequences of its own textual descriptions. Specifically, we selected the Qwen2.5-VL-7B (Bai et al., 2025) model for the VLM in the framework, maintaining consistency with the trained baseline, while the diffusion models in two stages are implemented using Qwen-Image (Wu et al., 2025a) and OmniGen2 (Wu et al., 2025b), respectively. In line with the closed-loop design, the VLM within the data synthesis module is dynamically updated from training checkpoints, enabling a co-evolution of the model and its training data. Based on our framework, we constructed a 10K perception-intensive vision-language dataset *Viper10K*, including 7K *Caption Self-Refining* data and 3K *Visual-Operation Predicting* data. The detailed implementation and instructions for the dataset construction are thoroughly presented in the Appendix A to ensure reproducibility and to facilitate future research within the community.

### 3.2.2 Two-stage Reinforcement Learning

To align with the progressive cognitive demands of the two-stage task, we designed a phased reinforcement learning approach, as shown in the lower part of Figure 1. Since all training data is self-synthesized by the model undergoing training, distribution shift from heterogeneous data sources is eliminated. This self-sourcing strategy renders the entire RL process free from any cold-start requirement. The training proceeds sequentially: the first phase utilizes the *Caption Self-Refining* data, followed by the second phase focused on the *Visual-Operation Predicting* task. A unified reward computation mechanism is applied consistently across both stages.

**Reward mechanism:** For each model output $O$, we split it into a set of text sequences and use the BGE-M3 model (Chen et al., 2024a) to calculate the semantic similarity matrix between these sequences and the ground truth. The reward is applied as follows:

$$R_{\text{format}} = \begin{cases} 1 & \text{if } O \text{ matches the expected format} \\ 0 & \text{otherwise} \end{cases}, \tag{5}$$

$$R_{\text{correct}} = \left( \frac{\sum_{i=1}^{N} \mathbb{1}\left[\max_{j=1}^{M} \text{sim}(s_i, g_j) \geq \tau\right]}{N} \right) \times \left( \frac{\sum_{i=1}^{N} \left( \mathbb{1}\left[\max_{j=1}^{M} \text{sim}(s_i, g_j) \geq \tau\right] \cdot L(s_i) \right)}{\sum_{i=1}^{N} L(s_i)} \right),$$
(6)

$$R = w_f \cdot R_{\text{format}} + w_c \cdot R_{\text{correct}},$$
(7)

Here, $G = \{g_1, g_2, \ldots, g_M\}$ is the ground truth set, $S = \{s_1, s_2, \ldots, s_N\}$ is the set of text sequences obtained by splitting the output $O$, $L(s_i)$ is the character length of the split sentence $s_i$, $\text{sim}(s_i, g_j)$ is the semantic similarity between the sentence $s_i$ and the ground truth $g_j$, and $\tau$ is the similarity threshold, set to 0.85. The weight coefficients $w_f$ and $w_c$ are set as $w_f = 0.05$ and $w_c = 0.95$.

**Optimization process:** We adopt a variation of Group Relative Policy Optimization (GRPO) algorithm, which has been proven effective in tasks involving mathematically structured reasoning chains (Ren et al., 2025). Following DAPO, we decouple the lower and higher clipping range to promote the diversity of the system as well as avoiding entropy collapse, and remove the KL penalty term to to encourage bolder policy updates. Specifically, for each input question $q$, we first sample a set of outputs $\{o_1, o_2, \ldots, o_n\}$, and then apply the reward function in Equation 7 to compute the reward score for each output. The policy model is optimized by maximizing the following objective:

$$\mathcal{J}(\theta) = \quad \mathbb{E}[q \sim P(Q), \{o_i\}_{i=1}^{G} \sim \pi_{\theta_{\text{old}}}(O|q)]$$
$$\frac{1}{G} \sum_{i=1}^{G} \frac{1}{|o_i|} \sum_{t=1}^{|o_i|} \left\{ \min\left[ r_{i,t}(\theta)\hat{A}_{i,t}, \text{clip}\left(r_{i,t}(\theta), 1 - \epsilon_{low}, 1 + \epsilon_{high}\right) \hat{A}_{i,t} \right] \right\},$$
(8)

where,

$$r_{i,t}(\theta) = \frac{\pi_\theta(o_{i,t} \mid q, o_{i,<t})}{\pi_{\theta_{\text{old}}}(o_{i,t} \mid q, o_{i,<t})}, \quad \hat{A}_{i,t} = \frac{R_i - \text{mean}(\{R_i\}_{i=1}^{G})}{\text{std}(\{R_i\}_{i=1}^{G})}.$$
(9)

## 4 EXPERIMENTS

### 4.1 EXPERIMENTAL SETUP

We selected Qwen2.5-VL-3B and Qwen2.5-VL-7B as base models and performed two-stage RL training on *Viper10K*, resulting in Qwen-Viper series. Identical hyperparameters were used for both stages: a batch size of 128, 5 rollouts per prompt at temperature 1.0, and rewards computed by the BGE-M3 model following the Equation 7. More details are presented in Appendix A.1.

### 4.2 MAIN RESULTS

#### 4.2.1 BENCHMARKS

In our work, we use seven multimodal benchmarks for evaluation: (1) **MMStar** (Chen et al., 2024b) emphasizes visual dependency, ensuring that answer inference requires visual grounding while minimizing potential data leakage. (2) **RealWorldQA** (X.AI, 2024) focuses on image understanding and verifiable reasoning in real-world scenarios. (3) **MME-RW(en)** (Zhang et al., 2025e) is the English subset of MME-RW, improving over previous benchmarks in scale, resolution, and task complexity for real-world applications. (4) **BLINK** (Fu et al., 2024) examines core visual perception abilities such as depth estimation, visual correspondence, and multi-view reasoning. (5) **MANTIS Eval** (Jiang et al., 2024) is designed to assesse multi-image reasoning, including reference resolution, comparison, and temporal understanding. (6) **HallusionBench** (Guan et al., 2023) evaluates model robustness against visual and language hallucinations. (7) **CRPE (relation)** (Wang et al., 2024b) evaluates models on subject–predicate–object structures, requiring recognition of both entities and their relations. The statics of each benchmark are detailed in the Appendix B.

#### 4.2.2 EXPERIMENTAL RESULTS

We evaluated our method on seven diverse vision-language benchmarks encompassing single-image, multi-image, and hallucination tasks. The selected benchmarks include both general and reality

Table 1: Experimental results on multiple benchmarks emcompassing single-image, multi-image and hallucination tasks, $\Delta \uparrow$ denotes the absolute gain of ViPER over the base model.

| Model | MMStar | RealWorldQA | MME-RW (en) | BLINK (val) | Mantis Eval | Hallucination Bench(Avg) | CRPE (relation) | Overall |
|---|---|---|---|---|---|---|---|---|
| GPT-4o-0513(Hurst et al., 2024) | 64.7 | 75.4 | 45.2 | 68.0 | – | 55.0 | 76.6 | – |
| LLaVA-OneVision-0.5B(Li et al., 2024) | 37.7 | 55.6 | – | 52.1 | 39.6 | 27.9 | – | – |
| InternVL2.5-1B(Chen et al., 2024c) | 50.1 | 57.5 | 44.2 | 42.0 | 51.2 | 39.0 | 60.9 | 49.3 |
| InternVL3-1B(Zhu et al., 2025) | 51.5 | 58.2 | 46.0 | 42.9 | 50.2 | 41.4 | 64.0 | 50.6 |
| InternVL2.5-2B(Chen et al., 2024c) | 53.7 | 60.1 | 48.8 | 44.0 | 54.8 | 42.6 | 70.2 | 53.5 |
| InternVL3-2B(Zhu et al., 2025) | 60.7 | 64.3 | 53.8 | 50.3 | 65.9 | 42.5 | 71.5 | 58.4 |
| Qwen2.5-VL-3B(Bai et al., 2025) | 55.9 | 65.4 | 53.1 | 47.6 | 68.7 | 46.3 | 73.6 | 58.7 |
| Qwen-Viper-3B | 57.8 | 67.7 | 54.6 | 49.2 | 70.0 | 49.1 | 74.3 | 60.4 |
| $\Delta \uparrow$ | 1.9 | 2.3 | 1.5 | 1.6 | 1.3 | 2.8 | 0.7 | 1.7 |
| MiniCPM-V2.6(Yao et al., 2024) | 57.5 | 65.0 | – | 53.0 | 69.0 | 48.1 | 75.2 | – |
| LLaVA-OneVison-7B(Li et al., 2024) | 61.7 | 66.3 | – | 48.2 | 64.2 | 46.8 | – | – |
| InternVL2.5-8B(Chen et al., 2024c) | 62.8 | 70.1 | 59.1 | 54.8 | 67.7 | 50.1 | 78.4 | 63.3 |
| InternVL3-8B(Zhu et al., 2025) | 68.2 | 70.8 | 62.0 | 55.5 | 70.1 | 49.9 | 76.3 | 64.7 |
| Qwen2.5-VL-7B(Bai et al., 2025) | 63.9 | 68.5 | 57.4 | 56.4 | 75.1 | 52.9 | 76.4 | 64.4 |
| +Viper10K (SFT) | 64.5 | 68.2 | 57.9 | 56.8 | 75.6 | 53.2 | 76.1 | 64.6 |
| Qwen-Viper-7B | 66.2 | 71.4 | 59.0 | 57.6 | 75.6 | 54.4 | 77.6 | 66.0 |
| $\Delta \uparrow$ | 2.3 | 2.9 | 1.6 | 1.2 | 0.5 | 1.5 | 1.2 | 1.6 |

Table 2: Performances across different domains. $\Delta \uparrow$ denotes the absolute gain of ViPER over the base model.

| Model | Coarse Perception | Fine-grained Perception | Instance Reasoning | Logic Reasoning | Math | Science & Technology |
|---|---|---|---|---|---|---|
| Qwen2.5-VL-3B | 68.8 | 48.4 | 62.4 | 56.4 | 62.0 | 37.2 |
| Qwen-Viper-3B | 70.0 | 52.8 | 64.4 | 57.6 | 62.8 | 39.2 |
| $\Delta \uparrow$ | 1.2 | 4.4 | 2.0 | 1.2 | 0.8 | 2.0 |
| Qwen2.5-VL-7B | 73.6 | 55.6 | 73.2 | 69.6 | 66.8 | 44.4 |
| Qwen-Viper-7B | 75.2 | 61.6 | 74.8 | 70.4 | 67.2 | 48.0 |
| $\Delta \uparrow$ | 1.6 | 6.0 | 1.6 | 0.8 | 0.4 | 3.6 |

perception-oriented visual question answering (VQA) tasks, aligning with our method's focus. As shown in Table 1, ViPER yielded average performance gains of 1.7% and 1.6% across all benchmarks for the 3B and 7B models, respectively.

To analyze the specific capability improvements, we conducted fine-grained evaluations across six subdomains, and results are shown in Table 2.[2] The improvements were most pronounced in Fine-grained Perception, where Qwen-Viper-3B and Qwen-Viper-7B achieved gains of 4.4% and 6.0% respectively, underscoring ViPER's primary effect on detailed visual understanding(e.g., object location/counting, attribute recognition). Stable gains were also observed in Coarse Perception and Instance Reasoning, confirming broader perceptual strengthening. Notably, despite no direct training on knowledge tasks, a significant 3.6% average improvement emerged in the Science & Technology domain for the 7B model. This compellingly suggests that enhanced perception enables more accurate integration of visual cues with parametric knowledge, demonstrating that model capabilities are deeply intertwined rather than modular.

Results on hallucination benchmarks further support this observation. Compared to the base models, Qwen-Viper exhibited lower hallucination rates, indicating that enhanced visual perception enables more faithful processing of image information and partially mitigates the negative impact of linguistic priors. Furthermore, performance gains on multi-image benchmarks confirm that the perception enhancements are fundamental and generalizable to out-of-domain tasks, enabling more effective understanding and reasoning across complex visual contexts.

---

[2]The data of all subdomains are from MMStar.

## 4.3 IN-DEPTH ANALYSIS

### 4.3.1 INFLUENCE OF COLD-START

Conventional RL methods often rely on high-quality cold-start data (Ren et al., 2025; Lee et al., 2024; Yu et al., 2025b). Integrating a cold-start phase with RL training typically leads to more stable and significant improvements compared to direct RL fine-tuning. However, owing to the self-bootstrapping nature of the ViPER method, the data used for RL training is entirely self-generated, thereby eliminating the distribution discrepancy typically introduced by external models.

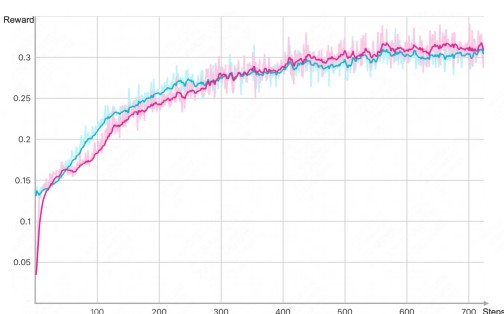

We conducted two experimental setups: a three-stage supervised SFT + RL process that includes a cold-start phase and a direct two-stage RL process without cold-start. In the cold-start stage, we used the Gemini-2.5-Pro model to generate 1K chain-of-thought data samples containing caption refinement and visual operation prediction annotations, which were then used to SFT of Qwen2.5-VL.

As shown in Figure 2, although the reward curve of the training process without cold-start began at a significantly lower level compared to that with cold-start, its reward growth trend

Figure 2: Reward curves with/without cold-start.

caught up with and surpassed the latter after 300 steps, eventually converging to a marginally higher final reward. The experimental phenomenon indicates that incorporating a cold-start phase in ViPER does not enhance model performance and may even constrain its potential for exploration and self-evolution. Thus, the proposed ViPER method effectively eliminates the dependency on high-quality cold-start data.

### 4.3.2 TWO-STAGE RL VS. MIXED RL

We further analyze the advantages of the two-stage RL strategy. A common practice in reinforcement learning is to mix data from different types of tasks during training, typically to prevent the model from overfitting to a specific task and suffering from performance degradation on others.

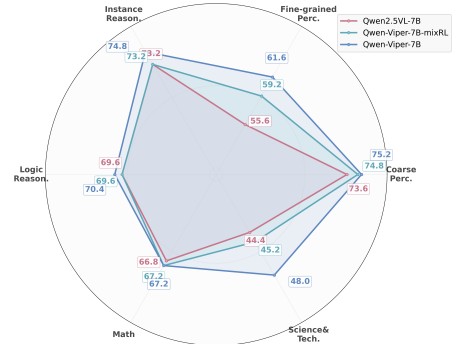

Our two-stage task follows a coarse-to-fine progressive process: the *Caption Self-Refining* stage focuses on global image information processing and emphasizes the understanding of static scenes, while the *Visual-Operation Predicting* stage targets more fine-grained visual cues and highlights the comprehension of dynamic changes. Therefore, a multi-stage RL approach that incrementally enhances the model presents an intuitively sound training paradigm.

To evaluate the training strategy, we compared our two-stage RL approach against a mixed RL process, where data from both stages were randomly blended during training on the *Viper10K* dataset. As illustrated in Figure 3, the two-stage RL strategy significantly outperforms the mixed training approach, demonstrating the necessity of the proposed phased training strategy.

Figure 3: Improvements of the two-stage RL vs. the mixed RL across six domains. The two-stage strategy outperforms the mixed RL.

To further investigate the effects before and after the two-stage training, we conducted a more in-depth analysis. For the *Caption Self-Refining* stage, we intriguingly observed the emergence of visual thinking during training. We visualized the word cloud of the chain-of-thought tokens after training, as shown in Figure 4a. The trained model spontaneously produced high-frequency operation verbs such as "scan" "zoom in" "look closely at" and "focus on", demonstrating a "thinking-with-images" reasoning pattern. For the *Visual-Operation Predicting* stage, we tracked changes in the model's attention distribution for identical visual input before and after training to analyze the

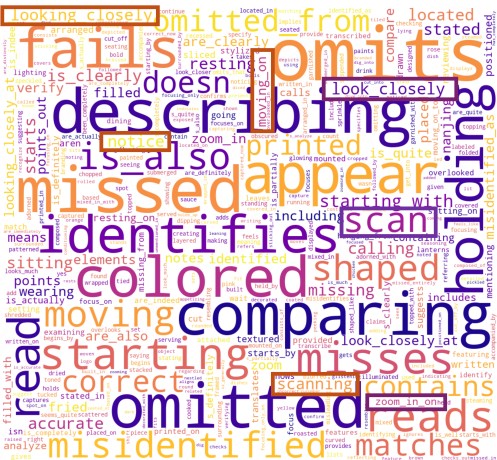

(a) Word cloud of chain-of-thought tokens from the model after *Caption Self-Refining* RL training. Verbs indicating visual operations are highlighted with colored boxes.

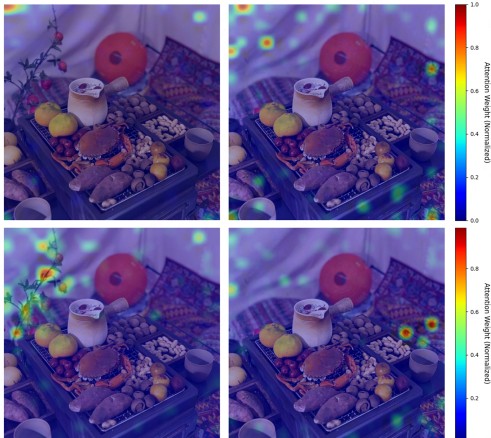

(b) Attention heatmaps for the same set of input images. The top two panels show the attention heatmaps from Qwen2.5-VL-7B, while the bottom corresponds to those from Qwen-Viper-7B.

Figure 4: Visualization of word cloud and attention.

impact of this training stage through the lens of attention mechanisms. The results are presented in Figure 4b. The model after the second stage showed significantly increased concentration on critical information, effectively translating the image operations described in the first-stage reasoning chains into shifts in attention patterns. This shift enables the model to spontaneously focus on and perform detailed analysis of local regions.

### 4.3.3 ABLATION STUDY

To validate the necessity of each stage, we conducted ablation studies. Specifically, we performed RL training solely with the *Caption Self-Refining* stage and separately with only the *Visual-Operation Predicting* stage, then compared their results against the fully trained Qwen-Viper model that underwent both stages. The detailed experimental outcomes are presented in Figure 5.

The ablation results confirm that while each stage contributes uniquely, their integration is key to significant gains. Training solely on the *Caption Self-Refining* stage fosters a foundational, global visual reasoning capability. By learning to generate and critique holistic descriptions, the model builds a robust scaffold for scene understanding, leading to balanced but moderate gains. In contrast, exclusive training on the *Visual-Operation Predicting* drives localized perceptual sensibility, which forces the model to master fine-grained attribute and relationship analysis, resulting in sharper improvements on corresponding tasks. Crucially,

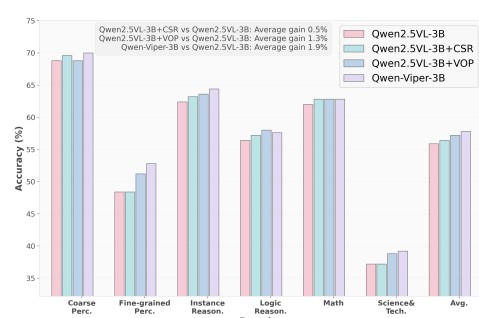

Figure 5: Ablation results on *Caption Self-Refining* and *Visual-Operation Predicting* Tasks.

the full framework's superiority stems from a coordinated mechanism: the first stage establishes reliable global context, which in turn primes and grounds the intensive local analysis of the second stage. This coherent progression from global to local enables a more profound and integrated self-evolution than either stage could achieve in isolation.

### 4.3.4 MECHANISMS FOR MITIGATING NOISES FROM GENERATIVE MODELS

ViPER's framework centrally leverages diffusion models, which are employed not only for caption-to-image generation but also for highly challenging fine-grained visual editing tasks. This reliance

imposes stringent requirements on the precision of diffusion models and consequently establishes a critical performance bottleneck. Current multimodal generative models often exhibit limitations in instruction-following capabilities, particularly in fine-grained compositional generation tasks, which may lead to data collapse and compromise training stability. To address the potential data noise introduced by generative models, we implement a dual-mechanism strategy:

First, for fine-grained editing tasks, we introduce VLM-as-Judge to filter generated outputs that fail to meet specified requirements. Implementation details and corresponding prompts are documented in Appendix A.4. Nevertheless, while VLM-as-Judge effectively ensures semantic fidelity, it provides inadequate control over non-semantic attributes such as lighting conditions and textural patterns. As a result, reconstructed images often inherit style transfer effects and textural artifacts characteristic of the generative model. To circumvent the impact of such generator-specific artifacts, we strategically design the reward function in the reinforcement learning phase to neutralize their influence. Although the VLM may occasionally produce self-critical content containing hallucinatory refinements due to biases introduced during data construction, it will never generate the same hallucinatory content during training because of the absence of reconstructed images. The reward of RL focuses exclusively on recall based on semantic similarity. Consequently, these minimally hallucinated refinements remain unmatched by the model's rollouts, thereby yielding zero advantage estimates and no gradient updates. This design systematically addresses the impact of data noise on the VLM's training process.

## 5 CONCLUSION

To break through the bottleneck of VLMs in perception-intensive tasks, we introduce a self-evolutionary framework ViPER, which establishes a closed-loop cycle of data construction and reinforcement fine-tuning. ViPER is built around a novel two-stage perceptual task that leverages image-level reconstruction and instance-level reconstruction to bootstrap visual perception and comprehension. This framework addresses the scarcity of high-quality perceptual data by enabling the model to autonomously generate its own training samples, while the subsequent post-training phase continuously refines the model's capabilities using the self-synthesized data. In this way, ViPER facilitates VLMs' self-evolutionary enhancement of visual perception, moving beyond the limitations of text-centric reasoning approaches. Extensive experiments demonstrate the effectiveness of ViPER and provide insights into its self-consistent design, strongly validating the role of generation in advancing perceptual understanding.

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

## A    IMPLEMENTATION DETAILS

### A.1    EXPERIMENTAL SETUP

We implemented the two-stage reinforcement learning based on the verl 0.3.1 [3]. The entire training process was conducted on a single node equipped with eight A100 80GB GPUs, leveraging Fully Sharded Data Parallel (FSDP) for efficient large-scale training. Key hyperparameters used in the reinforcement learning process are summarized in Table 3.

Table 3: Key experimental settings for the RL process

| Module | Parameter | Value |
|--------|-----------|-------|
| data | batch_size | 128 |
|  | max_prompt_length | 10240 |
|  | max_response_length | 4096 |
| actor | optim.lr | $1 \times 10^{-6}$ |
|  | mini_batch_size | 64 |
|  | micro_batch_size_per_gpu | 4 |
|  | use_kl_loss | False |
| sampling | temperature | 1.0 |
|  | rollout.n | 5 |
| hardware | num_gpus | 8 |
|  | memory_per_gpu | 80 GB |

### A.2    HEURISTIC RULES OF VISUAL OPERATIONS

For the data construction process of the visual-operation predicting task, we designed heuristic rules for the VLM. First, we predefined four visual operation tasks as toolkits, specifying their applicable scenarios and rules. These correspond to four common error types in fine-grained perception scenarios for VLMs: 1. Omission of small-scale details; 2. Confusion of spatial relationships; 3. Overemphasis on primary subjects while neglecting contextual backgrounds; 4. Incorrect fine-grained attribute discrimination;

The model selects the most suitable editing task upon thorough and meticulous analysis of the input image:

---

**Prompt for Heuristic Rules of Visual Operations**

**Role and Core Principle**
You are a creative and insightful visual analyst. Your task is to analyze the provided image and determine the most suitable category of subtle edit for creating challenging VQA (Visual Question Answering) data.
The edit category you choose must be the most suitable to the content of the image, and support fine-grained changes that require careful observation to notice.

**Task Description**
First, conduct a thorough analysis of the provided image. Describe all the details.
Next, refer to the "Editing Toolkit" below.
Based on your analysis, select the most suitable task category from the toolkit.
Finally, articulate your reasoning for this choice, explaining why this category is superior to others for creating a subtle and effective edit on this specific image.

**Editing Toolkit**

---

[3]https://verl.readthedocs.io/en/v0.3.x/start/install.html

- **Removing/Adding Details:** If the image contains many elements or objects with complex ingredients, consider removing or adding one of the small-scale objects.
- **Changing Spatial Relationships:** If an entity has a well-defined spatial relation to its surroundings and can be feasibly repositioned, then alter its location to disrupt and reconstruct the spatial context.
- **Editing Background:** If the image is clearly separated into a foreground and background, and the background is rich with elements, consider making edits to an object there.
- **Attribute Tuning:** If the image contains an object with a property that is easy to change, and the change would be subtle enough to require close inspection to notice, consider editing one of the object's properties.

**Output Format**
Your output MUST follow the JSON structure below. Do not include any other commentary or introductory text.
{
Image Analysis: [Your detailed analysis of the image in natural language.]
Edit Task: [The name of the task category you chose from the toolkit.]
Reasoning: [Your detailed justification for why this task category is the most suitable for this image, explaining how it enables a subtle and effective edit and why it is superior to other options.]
}

For the given visual operation tasks, we provide advanced and detailed task descriptions, and require the model to generate precise visual-operation instructions that are optimally tailored to the image content.

## A.3 Prompt in Dual-Level Reconstruction

We provide the key prompts used for dual-level reconstruction during the data synthesis process below to ensure full reproducibility of the pipeline:

**Prompt for Caption Self-Refining**

You are an expert AI assistant specializing in high-fidelity image caption evaluation.
Your task is to analyze an original image, its caption, and a reconstructed image that was generated based only on that caption. Your goal is to identify how the caption fails to accurately and completely describe the original image.
The core of your analysis is to compare the original image with the reconstructed image. The differences between these two images are a direct result of inaccuracies or omissions in the caption. For every significant difference you find, you must describe the corresponding flaw in the caption.

**Analysis Criteria:**
Analyze the caption's flaws based on the visual discrepancies between the original and reconstructed images:

- Omitted Details: Does the reconstructed image lack important objects, elements, or background details that are present in the original image? This indicates the caption omitted these details.

- Misidentified Entities: Are objects, people, or scenes in the reconstructed image fundamentally different from the original? (e.g., a dog instead of a cat, a city instead of a forest). This indicates the caption misidentified something.

- Inaccurate Spatial Relationships: Is the position or arrangement of elements in the reconstructed image incorrect compared to the original? This indicates the caption described spatial relationships inaccurately.

- Unmentioned or Incorrect Text: Is there text visible in the original image that is missing or garbled in the reconstructed image? This indicates the caption either omitted or incorrectly transcribed the text.

**Input:**

- Original Image: The ground-truth image
- Reconstructed Image: The image generated from the caption
- Current Caption: The text caption being evaluated

**Output Requirements:**

- Your response must be a single JSON object. This object must contain a single key, "refinement", which is a list of strings. Each string in the list must be a concise and precise description of a single factual error or significant omission you identified in the caption.
- Focus exclusively on factual errors and omissions that cause visual discrepancies.
- Crucially, do not describe the differences between the images in your output. Instead, describe the flaw in the caption that caused the difference.
- Do NOT comment on writing style, phrasing, tone, or subjective quality.
- If the caption is perfectly accurate and the reconstructed image is a faithful representation of the original, return a JSON object with an empty list: "refinement": [].

---

**Prompt for Generating Refined Captions**

**Role**
You are an extremely meticulous and precise text editor.

**Task**
Your sole task is to revise an "Original Caption" based on a list of "Refinement Suggestions". Your goal is to produce a fluent, coherent "Refined Caption" that fully incorporates all suggestions.

**Rules**

- **Must Integrate All Suggestions**: You must accurately reflect every point from the "Refinement Suggestions" in the new caption.
- **No New Information**: You are strictly forbidden from adding any new details that are not present in either the original caption or the suggestions.
- **Preserve Correct Information**: All accurate information from the original caption that is not targeted by a suggestion must be retained.
- **Single Text Output**: The final result must be a single, flowing descriptive paragraph, not a list of changes.
- **Strict Output Format**: Your response must contain only the final refined caption text. Do not include any preambles (e.g., "Here is the refined caption:"), explanations, or Markdown formatting.

**Input Data**

- **Original Caption**: {original_caption}
- **Refinement Suggestions**: {suggestions_str}

Please directly output the refined caption:

---

**Prompt for Generating Visual Editing Instructions**

You are a professional image editing instruction generator. Follow these steps:

1. First, carefully observe the image and identify all entities and elements in the image.
2. Analyze the cognitive difficulty of each entity (considering complexity, abstraction level, recognition difficulty, etc.).
3. Determine the entity with the highest cognitive difficulty and suitable to edit.
4. Based on the given task description and specific image content, generate a targeted editing instruction.

**Return JSON format with these fields:**

- "entity_to_edit": the entity chosen to be remove
- "editing_instruction": Comprehensive editing instruction (see requirements below)

**Requirements for editing instructions:**

- Must be concise, clear, and directly usable as a prompt for image editing/generation models
- Must comply with the given task description
- Avoid lengthy explanations, focus on direct editing commands

**Task Description:**
{task_description}

Please analyze this image and generate corresponding editing instructions.

---

## A.4   DATA QUALITY ASSURANCE

Precise and diverse visual editing instructions pose significant challenges for current open-source image editing models, which are prone to uncontrollable errors in fine-grained editing tasks. To address this, we have introduced a dual-validation mechanism combining CLIP scores and LLM-as-a-judge. The CLIP score serves as an initial filter to eliminate editing results that deviate significantly from the original image, while the LLM-as-a-judge employs stricter validation rules to ensure that the final edits strictly adhere to the instructions while maintaining high consistency with the original image.

---

**Prompt for Checking Instance-Level Reconstruction**

**Role**
You are an image validation specialist, and you specialize in rigorously assessing edited images to ensure they precisely adhere to the given instructions.

**The original image**
{image1}

**The edited image**
{image2}

Please evaluate the edited image compared to the original image, focusing only on semantic changes.

**Editing instruction**
{edit instruction}

**What to evaluate:**

---

- **Strict Adherence:** Does the edit accurately reflect the user's instruction in terms of content and objects?
- **No Unintended Semantic Changes:** Are there any new objects, removed objects (that were not supposed to be removed), or changes in the attributes of existing objects that were not part of the instruction?

**What to ignore:**
Minor, global variations in brightness, contrast, or color saturation. Consider these acceptable side effects unless the instruction was specifically about changing them.

**Decision:**
- If the edit is semantically accurate and clean, set "is_valid" to true.
- If the edit fails to follow the instruction or introduces unintended semantic changes, set "is_valid" to false and provide a brief explanation in the "reason" field.

Please return your judgment only in JSON format as follows:
{ "is_valid": boolean, "reason": "A brief explanation if is_valid is false, otherwise this field can be omitted." }

## B   BENCHMARK DETAILS

To provide a clearer understanding of the datasets used in our experiments, we present more detailed descriptions of the seven multimodal benchmarks:

(1) **MMStar** (Chen et al., 2024b) is designed to address the limitations of weak visual dependency and unintentional data leakage in previous benchmarks. It contains 1,500 human-curated samples that require genuine visual reasoning and are resistant to memorization effects. The benchmark spans six core capabilities and 18 fine-grained axes, ensuring a balanced and rigorous evaluation of large VLMs.

(2) **RealWorldQA** (X.AI, 2024) provides more than 700 images from real-world scenarios, including those captured from vehicles and other everyday contexts. Each image is paired with a question and a verifiable answer, making it a concise yet practical dataset to assess real-world grounding and reasoning in multimodal models.

(3) **MME-RW** (Zhang et al., 2025e) is currently one of the largest manually annotated multimodal benchmarks, comprising tens of thousands of high-quality images collected from public datasets and the Internet. Expert annotators generated diverse question–answer pairs covering challenging real-world subtasks. Compared with previous datasets, it features higher-resolution images and significantly greater task diversity. **MME-RW (en)** denotes its English subset, where all questions and answers are provided in English.

(4) **BLINK** (Fu et al., 2024) reformulates 14 classic computer vision tasks into 3,807 multiple-choice questions, paired with single or multiple images. The benchmark assesses core perceptual abilities, including relative depth estimation, visual correspondence, and multi-view reasoning.

(5) **Mantis Eval** (Jiang et al., 2024) is a benchmark specifically designed to evaluate multi-image reasoning in multi-modal models. It consists of carefully curated test cases that require reasoning across multiple images, including tasks such as reference resolution, cross-image comparison, and temporal understanding. By focusing on multi-image contexts, the benchmark provides a systematic way to assess whether models can integrate information beyond single-image perception.

(6) **HallusionBench** (Guan et al., 2023) is built to evaluate multimodal reasoning under hallucination-inducing conditions. It contains 346 images paired with 1,129 expert-designed questions, many organized into control structures that allow fine-grained analysis of logical consistency and error types.

(7) **CRPE** (Wang et al., 2024b) provides a structured evaluation of object recognition and relation comprehension. The benchmark consists of single-choice questions divided into four subsets: Existence, Subject, Predicate, and Object. This division enables systematic testing of whether models can detect the presence of objects, identify entities, and capture subject-predicate-object relations, offering insight into fine-grained relational reasoning abilities. In our experiments, we specifically adopt the **relation** subset to evaluate the subject-predicate-object reasoning.

## C  DATA SAMPLE

---

**Caption Self-Refining**

**Original Image & Reconstructed Image**

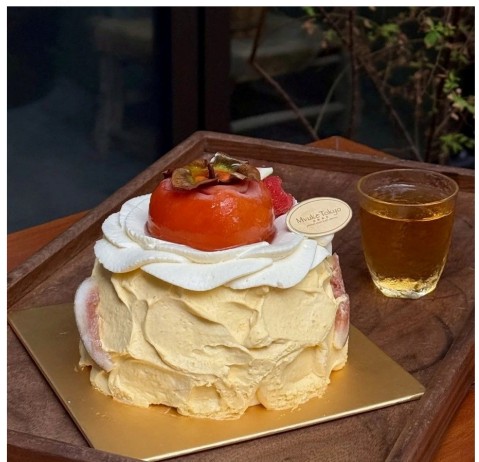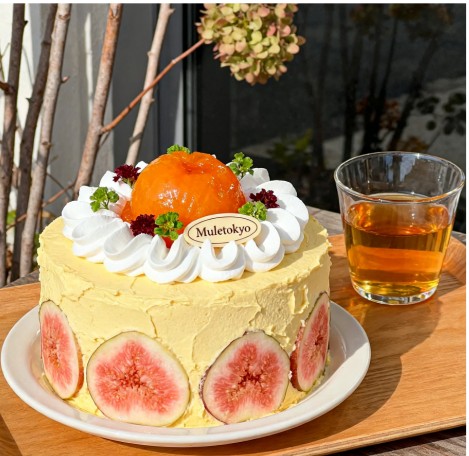

**Original Caption**

A decadent dessert featuring a round cake placed at the center of a wooden serving tray, embellished with intricate layers and vibrant toppings. The cake is covered in a rich, creamy frosting with a pale yellow hue displaying a rustic, textured finish reminiscent of hand-spread icing. Embedded along the sides are slices of ripe pink fruit, possibly figs, their interior pulp and outer skin visible through the frosting. Atop the cake sits an eye-catching glossy orange segment, likely a fruit compote or poached persimmon, crowned by small edible flower decorations with earthy green and burgundy tones. Surrounding the centerpiece is a white whipped cream spiral arranged in a flower-like shape, with a petite, oval-shaped decorative plaque reading "Muletokyo" nestled among the toppings. To the side of the cake, sitting on the same wooden tray, is a clear glass filled with amber-colored liquid, potentially tea or juice, offering a refreshing pairing to the indulgent treat. The background reveals an outdoor setting with hints of vegetation, characterized by thin branches and clusters of leaves, alongside a dark glass panel that reflects faint shapes and shadows.

**Refinement**

- The cake is placed on a square, gold-colored cake board, which is not mentioned in the caption.
- The 'small edible flower decorations' described on top of the orange fruit are the fruit's natural calyx, not separate decorations.
- The decorative plaque on the cake reads 'Myuke Tokyo', not 'Muletokyo'.
- A slice of a red fruit, possibly a raspberry, is visible on the whipped cream next to the persimmon, but is not mentioned.
- The cake is positioned on the left side of the wooden tray, not in the center.

**Visual-Operation Predicting**

**Original Image & Reconstructed Image**

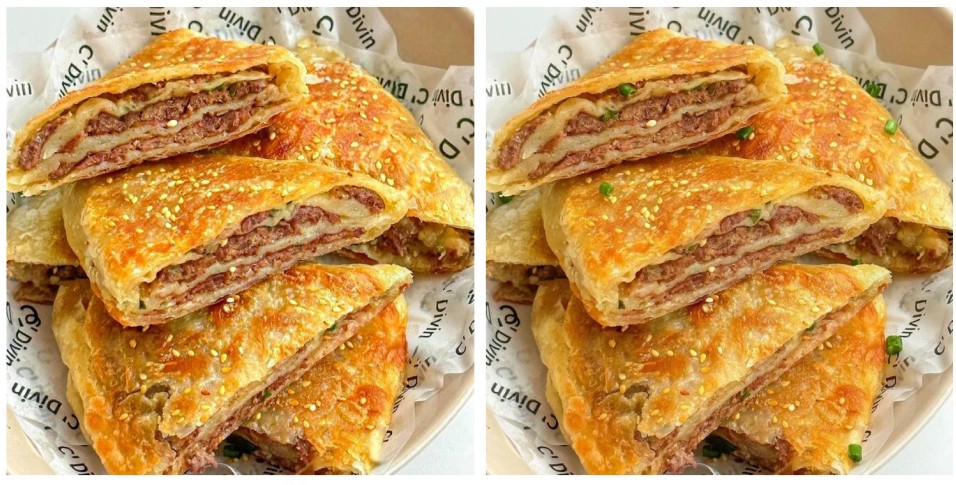

**Visual-Operation Instructions**
Add small chopped green scallions scattered on top of the sesame-covered meat pastries and on the paper lining in the basket.

## D  CASE STUDY

In this section, we present concrete cases to visually demonstrate the significant improvements achieved by our method. For each example, we provide responses from three models: Gemini-2.5-Pro (by API), Qwen2.5-VL-7B, and Qwen-Viper-7B. For the latter two models, we visualize their attention distributions during the response generation process, using attention heatmaps as a lens to observe the operational mechanism of the ViPER method.

---

**Case Study of Visual Counting**

**Question:** How many garages does the first building on our right have?

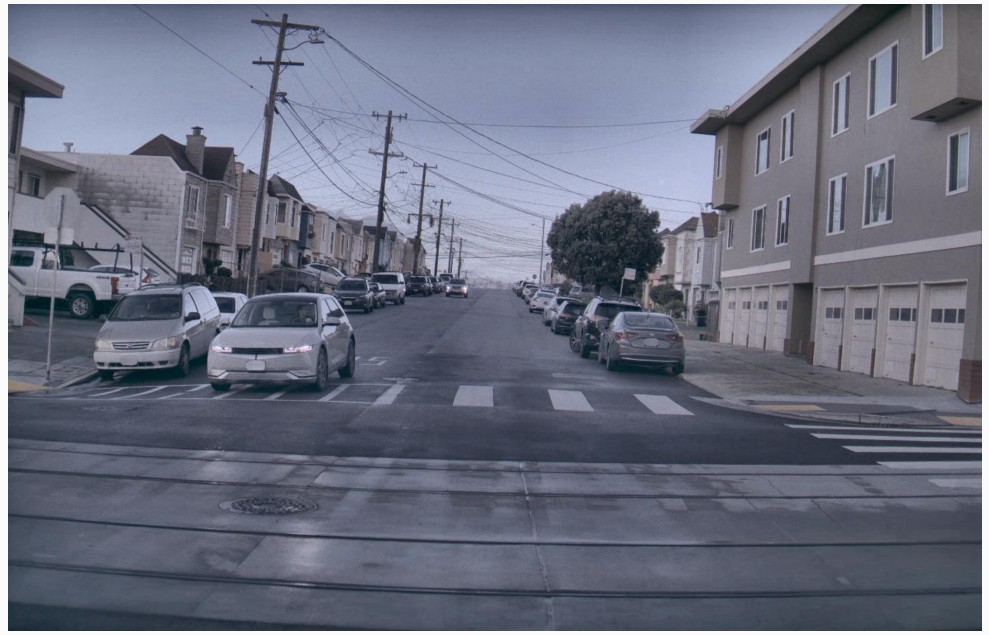

**Gemini-2.5-pro**
**Determining Garage Count**
I'm zeroing in on the user's query: specifically, how many garages are visible on the first structure to the right. My focus is on thoroughly examining the image to isolate and enumerate them.
**Focusing On Specifics**
Now, I'm focusing on the specifics. I've pinpointed the correct structure and need to accurately count the visible garages. Counting them carefully, I've identified five distinct garage doors on the ground floor. My next step is to formulate the answer concisely and confidently, incorporating a brief descriptive confirmation for clarity.
**Refining The Response**
Now, I'm taking a more analytical approach. I've broken down the user's question into distinct steps: identifying the building, counting the garages, and crafting a clear answer. I've confirmed that the target building is the most prominent on the right with five garage doors. I'm focusing on ensuring my response is both direct and detailed to instill trust.
Based on the image, the first building on the right has **five** garages.

---

**Qwen2.5-VL-7B**

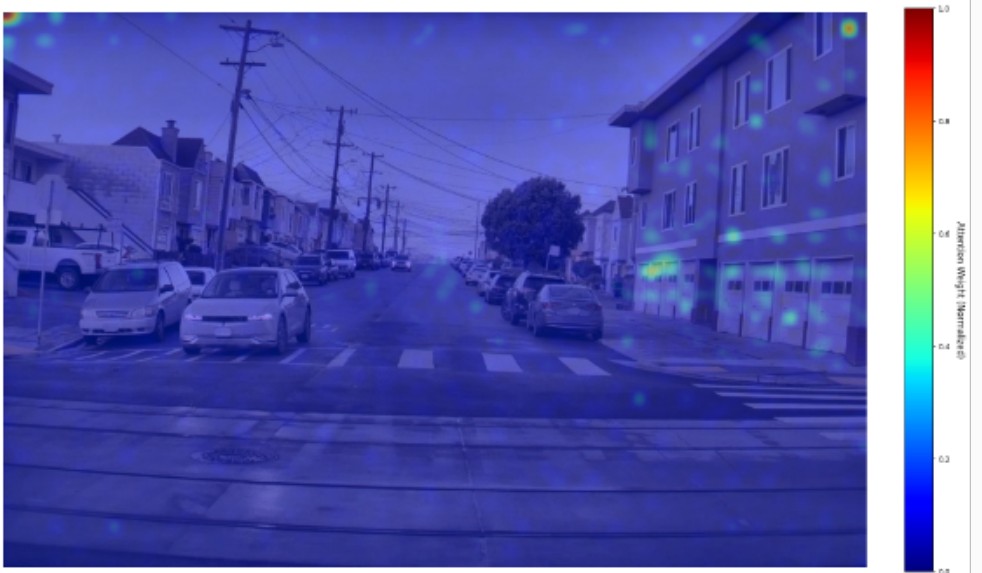

The first building on the right side of the image has **four** garage doors. These are clearly visible at the bottom of the building, just above the street level.

**Qwen-Viper-7B(Ours)**

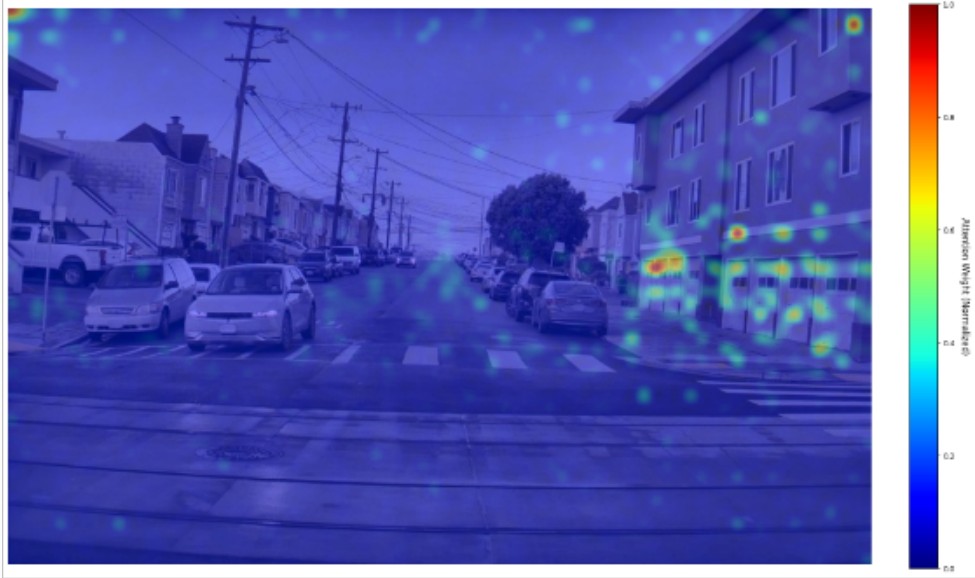

I'll start by examining the image, **focusing on** the structure on the right side of the street. The image is a view down a residential street with parked cars and buildings on both sides. My goal is to determine how many garages the first building on the right has.

I'm looking at this building, which appears to be a multi-story structure, likely an apartment complex or a similar type of building. It's a light grayish-brown color. As my eyes **scan its facade**, I see what looks like a row of white doors. These are clearly visible and are located directly beneath a row of windows on the ground floor. They are evenly spaced and appear to be identical in size and shape.

Next, I'll **do a quick visual count** of these doors. Starting from the left, I can see one, two, three, four, five, six, seven, and eight of them. They are all aligned in a straight line, making it a simple count. This building on the right has a total of eight white garage doors.

The question specifically asks for the number of garages on the first building on the right, and I can confidently state that there are **eight** of them.

---

**Case Study of Fine-grained Perception**

**Question:** What is the color scheme of the costumes worn by the men in the parade?
**Options:** A: Yellow, red, and gold, B: Red, blue, and black, C: Green, orange, and white, D: Black, white, and gray

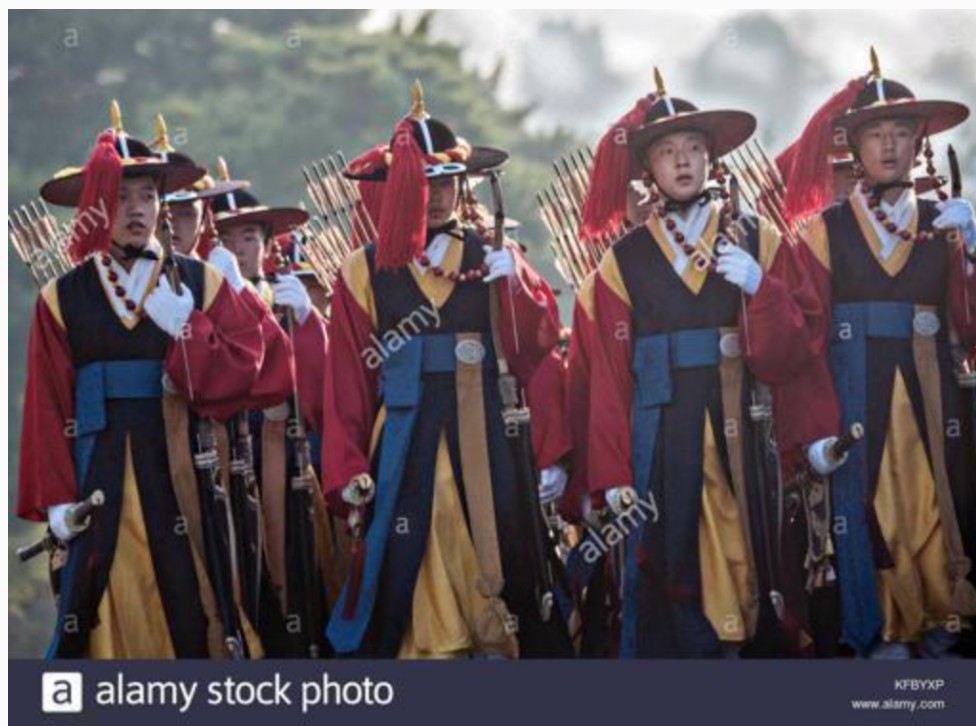

**Gemini-2.5-pro**

Based on the image, the primary colors visible in the costumes are:
Red (outer robes, tassels on hats)
Blue (sashes/belts)
Yellow/Gold (inner garments/trim)
Black (main body of the tunic/vest, hats)
White (gloves, neck trim)

Comparing this to the options:
A: Yellow, red, and gold (Missing prominent black and blue)
B: Red, blue, and black (These are all very prominent colors)
C: Green, orange, and white (Green and orange are not major colors)
D: Black, white, and gray (Missing prominent red, blue, and yellow)

The most accurately descriptive option focusing on the major, dominant colors is B: Red, blue, and black, as these three are clearly primary components of the main garment layers. However, since Yellow is also very prominent, let's look closer. The yellow is a significant color block. The options are a bit restrictive.

If we have to choose the best fit:
Red, blue, and black are all very strong, dominant colors.

Red, blue, and yellow/gold are also very strong, dominant colors.

Given the choices, B: Red, blue, and black captures three of the most contrasting and major colors in the main body of the uniform. However, if the yellow parts are considered "gold," A: Yellow, red, and gold would also be a strong contender, focusing on the brightest colors, but it misses the prominent black and blue.

Let's stick with the most prominent and contrasting large color blocks: Red, blue, and black appear to be the dominant features.

The correct option is B: Red, blue, and black.

**Qwen2.5-VL-7B**

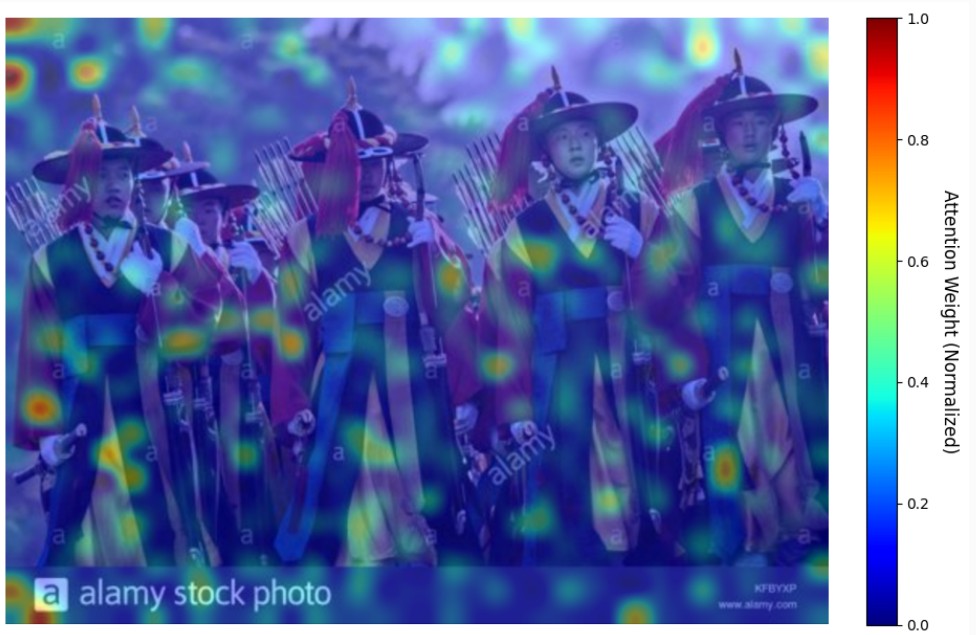

The color scheme of the costumes worn by the men in the parade is:

A: Yellow, red, and gold.

**Qwen-Viper-7B(Ours)**

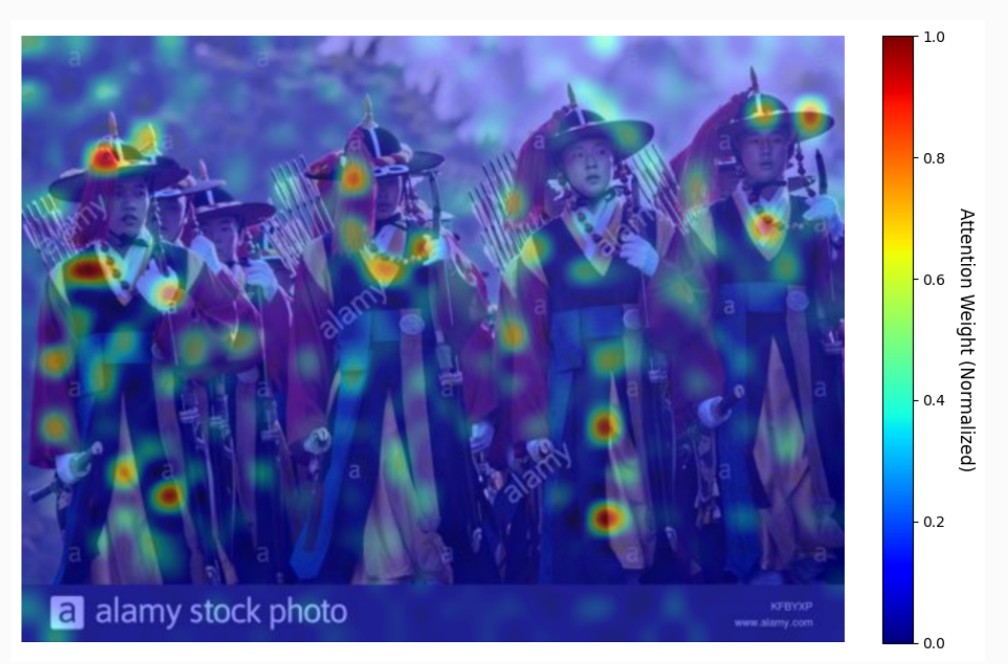

I'll start by examining the image of the group of men in what appears to be a traditional parade. My goal is to determine the color scheme of their costumes from the options provided.

First, I'll look at the central figure on the left. He's wearing a hat with a red tassel, a red and balck robe, and a blue belt. The next man to his right is wearing a similar outfit but with a red hat and a red and black top. As my eyes scan the group, I can see this pattern continues. The colors that stand out are red, black, and blue, along with some golden elements like the trims.

Next, I'll compare these colors against the given options. Option A mentions yellow, red, and gold, which misses the black and blue as main colors. Option B includes red, blue, and black, which is a perfect match for what I'm seeing. Option C lists green, orange, and white, none of which are present. And option D describes black, white, and gray, which are not part of the costumes at all.

After carefully reviewing the image, the most fitting description is clearly B: Red, blue, and black. That's my final answer.

## E   RESULTS ON VIRL39K

Table 4: Comparison between Qwen2.5-VL-7B and the version RFT on ViRL39K data, $\Delta$ denotes the variations in accuracy.

| Model | MMStar | RealWorldQA | MME-RW (en) | BLINK (val) | Mantis Eval | Hallusion Bench(Avg) | CRPE (relation) | Overall |
|---|---|---|---|---|---|---|---|---|
| Qwen2.5-VL-7B | 63.9 | 68.5 | 57.4 | 56.4 | 75.1 | 52.9 | 76.4 | 64.4 |
| Qwen2.5-VL-7BViRL39K | 63.4 | 68.1 | 57.1 | 57.0 | 74.2 | 53.7 | 76.8 | 64.3 |
| $\Delta$ | 0.5↓ | 0.4↓ | 0.3↓ | 0.6↑ | 0.9↓ | 0.8↑ | 0.4↑ | 0.1↓ |

To compare the training data synthesized by the ViPER framework with existing multimodal reasoning datasets, we fine-tuned Qwen2.5-VL-7B using the comprehensive ViRL39K dataset (Wang et al., 2025a) via the GRPO algorithm. ViRL39K rovides a curated collection of 38,870 verifiable QAs for Vision-Language RL training, with over 80% are math or chart/diagram reasoning

questions, The results summarized in Table 4 show that the fine-tuned model **did not** achieve superior performance on the evaluated benchmarks. Although modest improvements were observed on hallucination-oriented tasks, the model slightly underperformed the baseline on single-image and multi-image benchmarks emphasizing real-world perception. This outcome highlights two key insights: 1.The reasoning paradigms derived from mathematical and chart-oriented tasks exhibit **limited generalizability** to perception-intensive visual tasks; 2.It underscores the targeted efficacy of our proposed method in specifically enhancing visual perceptual capabilities.

## F   DATA AVAILABILITY STATEMENT

All image data employed in this study are under formal authorization for academic research purposes, ensuring full compliance with legal and ethical standards regarding copyright protection and privacy preservation.

