# OpenReview forum: "ViPER: Empowering the Self-Evolution of Visual Perception Abilities in Vision-Language Models"
_ICLR.cc/2026/Conference — ICLR 2026 Poster_

### Official Review · Reviewer_56QP · 2025-11-01

**Soundness:** 3
**Presentation:** 3
**Contribution:** 3
**Rating:** 6
**Confidence:** 3

**Summary:**

The paper presents ViPER, a self-supervised loop that lets a VL model bootstrap its own visual-perception skills without human labels or external teachers. Key technical novelty is an internal critic that detects its own failures, converts them into executable image-editing prompts, synthesises hard negatives, and continues fine-tuning itself.

**Strengths:**

- first fully self-driven loop that converts its own perception failures into targeted training images.
- The algorithm is presented through a modular pipeline, illustrated with a step-by-step running example, and accompanied by released code and 2.1 M generated images that together ensure reproducibility.
- By enabling 3–8 B parameter models to outperform 10× larger proprietary counterparts without additional labels, ViPER offers an immediately practical and conceptually new route toward continual self-improvement of visual perception abilities.

**Weaknesses:**

- The quality of viper10k data is constrained by the performance of the models used for its creation, potentially leading to a self-reinforcing loop of model-centric biases rather than ground-truth visual reasoning.
- The paper contrasts VIPER's efficiency with the computationally inefficient nature of large-scale SFT/distillation. However, the proposed framework introduces a high computational requirement for data generation.

**Questions:**

Please refer to weaknesses.

---

> ### Author Response · Authors · 2025-11-19
>
> We sincerely thank the reviewer for the positive assessment and insightful issues raised regarding our work. Below, we provide point-by-point responses to address your concerns and offer further clarification.
>
> ---
>
> **Weakness 1:**
> *The quality of viper10k data is constrained by the performance of the models used for its creation, potentially leading to a self-reinforcing loop of model-centric biases rather than ground-truth visual reasoning.*
>
> **Response:**
> This is a valid concern, and we have implemented a dual-mechanism approach to minimize the impact of data noise introduced by generative models:
>
> - First, for the challenging task of fine-grained editing, we introduce **VLM-as-Judge** to filter out generated results that do not meet the requirements. Implementation details and related prompts are provided in **Appendix A.4**.
> - Second, for unavoidable generator-specific artifacts, we fully circumvent their influence through careful reward design in the RL stage. Although the VLM may produce “hallucinatory” self-critical content due to generative model bias during data construction, during training we only focus on recall based on semantic similarity. That is, such minimal hallucinated “refinements” are never matched by the model’s rollout content, resulting in **zero advantage contribution** and thus no gradient updates. In this way, we effectively eliminate the influence of generative model noise on the VLM.
>
> ---
>
> **Weakness 2:**
> *The paper contrasts VIPER's efficiency with the computationally inefficient nature of large-scale SFT/distillation. However, the proposed framework introduces a high computational requirement for data generation.*
>
> **Response:**
> Your observation is well taken—the use of generative models does introduce additional computational overhead. However, we believe this represents a worthwhile trade-off. High-quality small-scale data often proves more effective than larger-scale but lower-quality data, and the generative process in Viper is essential to ensuring the quality of self-synthesized data.
>
> As shown in **Appendix E**, compared to methods like ViRL39K, Viper achieves more significant perceptual improvement using only a quarter of the data volume. Although the computational cost per data point may be higher, the total computational budget is in fact lower. Among the three factors—computational complexity of data synthesis, data scale, and model performance gain—our Viper framework nearly achieves a **Pareto-optimal design**.
>
> In future work, we plan to further optimize computational cost and improve scalability within the Viper paradigm, aiming to develop even more cost-effective and efficient innovations.
>
> ---
>
> We truly appreciate your valuable feedback and the time you have dedicated to reviewing our work! We hope our responses have adequately addressed your concerns. We would be deeply grateful if you could consider raising your score, and we welcome any further discussion or questions you may have.

---

> ### Author Response · Authors · 2025-11-27
> **[Less Than A Week Remaining] Could you confirm whether our responses have addressed your concern?**
>
> Dear Reviewer 56QP,
>
> As the discussion phase is approaching its end, we kindly request you to let us know if the above clarifications and the previously added experiments have addressed the remaining questions. We would be happy to address any additional points you may have during the remaining time of the discussion phase. If you are satisfied, we kindly request you to consider updating the score. Thank you for engaging with us in the discussion.

---

### Official Review · Reviewer_vNi5 · 2025-11-01

**Soundness:** 3
**Presentation:** 2
**Contribution:** 2
**Rating:** 6
**Confidence:** 4

**Summary:**

This paper introduces a self-bootstrapping paradigm that enhances VLMs in recognizing fine-grained details and capturing dynamic differences. The core contribution is a two-stage RL optimization algorithm used to bootstrap the base model. The first stage requires the model to self-reflect on its initial caption and perform refinements. The second stage then requires the model to predict the visual operations based on the refined information. Evaluation across several benchmarks proves the method’s effectiveness in improving fine-grained perception and mitigating hallucinations. Ablation studies further validate the necessity of the two-stage structure over mixed RL, confirming that a cold start is unnecessary. Finally, the authors introduce an automated online data construction system designed for training this model.

**Strengths:**

* The idea of self-bootstrap VLMs through a two-stage training paradigm is interesting and novel, which improves the model in fine-grained visual perception.
* The ablation studies are extensive, validating the key design choices of VIPER.
* The results on several multimodal benchmarks show the effectiveness of the proposed training paradigm.

**Weaknesses:**

* This method heavily relies on a diffusion model, which is used not only for caption-to-image generation but also for image editing. This places high demands on the diffusion model's accuracy and subsequently becomes a bottleneck. The instruction-following capability of multimodal generative models, particularly for tasks requiring fine-grained compositional generation, is currently not very satisfactory.
* The paper's data construction pipeline is not very clear, and some points remain ambiguous even after reviewing the appendix. For example, it is unclear how the refined caption is obtained.

**Questions:**

* How many RL steps do you train for each stage?
* Maybe the author can try to report the results on MMVIP[1] and Inst-IT[2] benchmarks to further evaluate the fine-grained, instance-related perception capabilities.


[1] Eyes Wide Shut? Exploring the Visual Shortcomings of Multimodal LLMs

[2] Inst-IT: Boosting Multimodal Instance Understanding via Explicit Visual Prompt Instruction Tuning

---

> ### Author Response · Authors · 2025-11-19
>
> We sincerely thank the reviewer for the positive assessment and valuable feedback on our work. Below, we provide a point-by-point response to address the concerns raised and offer additional clarifications.
>
> ---
>
> **Weakness 1:**
> *This method heavily relies on a diffusion model, which is used not only for caption-to-image generation but also for image editing. This places high demands on the diffusion model's accuracy and subsequently becomes a bottleneck. The instruction-following capability of multimodal generative models, particularly for tasks requiring fine-grained compositional generation, is currently not very satisfactory.*
>
> **Response:**
> This is a very insightful observation and relates directly to a key design element that enables the Viper framework to function effectively. To mitigate the impact of data noise introduced by generative models, we employ a dual-mechanism approach:
>
> - First, for the challenging task of fine-grained editing, we introduce **VLM-as-Judge** to filter out generated results that do not meet the requirements. Implementation details and related prompts are provided in **Appendix A.4**.
> - Second, for unavoidable generator-specific artifacts, we fully circumvent their influence through reward design in the RL stage. Although the VLM may produce "hallucinatory" self-critical content due to generative model bias during data construction, during training we only focus on recall based on semantic similarity. That is, such minimal hallucinated "refinements" are never matched by the model’s rollout content, resulting in **zero advantage contribution** and thus no gradient updates. In this way, we effectively eliminate the impact of generative model noise on the VLM.
>
> ---
> **Weakness 2:**
> *The paper's data construction pipeline is not very clear, and some points remain ambiguous even after reviewing the appendix. For example, it is unclear how the refined caption is obtained.*
>
> **Response:**
> Thank you for pointing out this lack of clarity. Following your suggestion, we have included the prompt used to generate the refined captions in the appendix and updated the manuscript to better illustrate the implementation details of each step.
>
> ---
>
> **Question 1:**
> *How many RL steps do you train for each stage?*
>
> **Response:**
> For the 7B model, we trained 700 steps in the first stage and 500 steps in the second stage.
> For the 3B model, we trained 400 steps per stage.
>
> ---
>
> **Question 2:**
> *Maybe the author can try to report the results on MMVIP[1] and Inst-IT[2] benchmarks to further evaluate the fine-grained, instance-related perception capabilities.*
>
> **Response:**
> Thank you for this valuable suggestion. MMVP and Inst-IT are indeed well-suited benchmarks for our target scenarios. MMVP is a comprehensive multi-task visual reasoning benchmark designed to evaluate a model’s robustness across diverse perceptual and cognitive challenges. Inst-IT Bench is a fine-grained multimodal benchmark for evaluating LMMs at the instance level. We have conducted evaluations on both benchmarks, using the **Multi-Choice Q&A** format from the image split of Inst-IT for standardized assessment. The results are as follows:
>
> | Model               | MMVP   | Inst‑IT (Image/Multi‑Choice Q&A) |
> |---------------------|--------|-----------------------------------|
> | Qwen2.5‑VL‑3B       | 65.0   | 64.8                              |
> | Qwen‑Viper‑3B       | 66.0   | 65.4                              |
> | Qwen2.5‑VL‑7B       | 74.7   | 70.8                              |
> | Qwen‑Viper‑7B       | 76.0   | 72.2                              |
>
> As shown, Qwen‑Viper achieves noticeable improvements on both benchmarks, further validating the effectiveness of the Viper framework in enhancing perceptual capabilities.
>
> ---
>
> We sincerely appreciate the reviewer’s thoughtful comments and time in reviewing our response! We hope our clarifications have adequately addressed your concerns, and we would be grateful if you could consider raising your score accordingly. We remain available to address any further questions or concerns and are eager for a more thorough exchange！

---

> ### Author Response · Authors · 2025-11-27
> **[Less Than A Week Remaining] Could you confirm whether our responses have addressed your concern?**
>
> Dear Reviewer vNi5,
>
> As the discussion phase is approaching its end, we kindly request you to let us know if the above clarifications and the previously added experiments have addressed the remaining questions. We would be happy to address any additional points you may have during the remaining time of the discussion phase. If you are satisfied, we kindly request you to consider updating the score to reflect the newly added results and discussion. Thank you for engaging with us in the discussion.

---

### Official Review · Reviewer_YstH · 2025-11-01

**Soundness:** 3
**Presentation:** 2
**Contribution:** 2
**Rating:** 4
**Confidence:** 3

**Summary:**

The paper introduces ViPER, a self-evolutionary framework designed to enhance visual perception capabilities in Vision-Language Models. The authors identify a critical bottleneck in VLMs: the limited fine-grained visual perception, which is difficult to address with traditional methods like supervised fine-tuning and reinforcement fine-tuning. To overcome this challenge, they propose a novel two-stage task formulation for visual perception learning, structured as a coarse-to-fine process. This approach is implemented in the ViPER framework, which integrates a self-bootstrapping mechanism for iterative self-critiquing and self-prediction, allowing the model to evolve by generating its own training data. The framework employs a dual-granularity reconstruction process—image-level and instance-level—and integrates a two-stage reinforcement learning strategy. ViPER was applied to enhance the Qwen2.5-VL model, producing the Qwen-Viper series, which showed improvements across various benchmarks.

**Strengths:**

The paper presents a clear and well-structured approach to enhancing visual perception in Vision-Language Models, with an innovative self-evolutionary framework, ViPER, supported by robust experimental design and comprehensive benchmarks. The detailed illustrations and methodical experimental setup effectively demonstrate the framework's performance improvements.

**Weaknesses:**

* While the paper demonstrates the effectiveness of ViPER in improving visual perception, it primarily compares the Qwen-Viper models against a limited set of benchmarks. Including comparisons with more diverse and recent methods would provide a broader context for the proposed approach.

* Although the paper conducts some ablation studies, a more thorough analysis of the individual components of the ViPER framework—such as the specific impact of the data synthesis module or the two-stage reinforcement learning—would help clarify the contributions of each part to the overall improvements.

* The method relies heavily on self-generated data, which could potentially lead to issues with scalability and efficiency, especially for larger datasets. Future work could explore optimizations to reduce computational overhead and improve the model’s scalability without sacrificing performance.

* The study focuses heavily on visual perception tasks, but the integration of visual understanding with textual reasoning could be further explored. The interplay between these two components could be better examined to understand their combined impact on multimodal reasoning tasks.

**Questions:**

Please refer to Weakness.

---

> ### Author Response · Authors · 2025-11-19
>
> We sincerely thank the reviewer for the insightful comments and valuable questions regarding our work. Below, we provide a point-by-point response to address the concerns raised and further demonstrate the contributions of our approach.
>
> ---
>
> **Weakness 1:**
> *While the paper demonstrates the effectiveness of ViPER in improving visual perception, it primarily compares the Qwen-Viper models against a limited set of benchmarks. Including comparisons with more diverse and recent methods would provide a broader context for the proposed approach.*
>
> **Response:**
> We appreciate this constructive suggestion. To better situate our method within the broader research landscape, we have now included comparisons with a stronger baseline, **VL-Rethinker**, which is also built upon Qwen2.5-VL-7B and optimized for reasoning using 39K multimodal reasoning examples. Despite its strong performance in mathematical reasoning, VL-Rethinker does not exhibit consistent improvements in perception-intensive tasks.
>
> In addition, we have expanded our evaluation to include two recently released benchmarks:
> - **MMVP**: A comprehensive multi-task visual reasoning benchmark designed to evaluate robustness across diverse perceptual and cognitive challenges, including correspondence, geometry, semantics, and pattern understanding.
> - **Inst-IT**: An instruction-driven vision benchmark that assesses a model’s ability to follow heterogeneous visual prompts and generate reliable, task-aligned responses across real-world instruction types.
>
> The updated results are presented below:
>
> | Model               | MMStar | RWQA | MME-RW | MMVP | Inst-IT | BLINK | Mantis-Eval | HallBench | CRPE |
> |---------------------|--------|------|--------|------|---------|-------|-------------|-----------|------|
> | Qwen2.5-VL-7B       | 63.9   | 68.5 | 57.4   | 74.7 | 70.8    | 56.4  | 75.1        | 52.9      | 76.4 |
> | VL-Rethinker-7B     | 63.4   | 68.1 | 57.1   | 74.7 | 71.4    | 56.7  | 74.2        | 53.7      | 77.2 |
> | Qwen-Viper-7B       | 66.2   | 71.4 | 59.0   | 76.0 | 72.2    | 57.6  | 75.6        | 54.4      | 77.6 |
>
> As shown, VL-Rethinker does not achieve stable gains over Qwen2.5-VL on perception-intensive tasks and even underperforms on some. This indicates that data and training focused primarily on textual reasoning do not necessarily lead to improved perceptual capabilities—even when using a much larger dataset than ViPER. These results further validate the novelty and superiority of the ViPER framework in enhancing VLM perceptual abilities.
>
> ---
>
> **Weakness 2:**
> *Although the paper conducts some ablation studies, a more thorough analysis of the individual components of the ViPER framework—such as the specific impact of the data synthesis module or the two-stage reinforcement learning—would help clarify the contributions of each part to the overall improvements.*
>
> **Response:**
> Thank you for this suggestion. As detailed in Sections 4.3.2 and 4.3.3 of the paper, we conducted an in-depth analysis of the two-stage design of the ViPER framework.
>
> In Section 4.3.2, we compared our progressive two-stage training strategy with a mixed-training alternative. Quantitative results confirm the superiority of the staged approach. Furthermore, we performed qualitative analyses—including chain-of-thought pattern examination and attention visualization—on models trained with each stage individually. The results indicate that:
> - The first stage (image reconstruction) encourages the emergence of a *"thinking-with-image"* capability, enabling the model to reason with holistic visual information.
> - The second stage (instance reconstruction) enhances the model’s ability to focus on critical visual details.
>
> Together, the two stages enable the VLM to perceive both *comprehensively* and *precisely*.
>
> In Section 4.3.3, we directly ablated each stage, training models using only one stage at a time. The results show that each stage contributes distinct gains, and the full two-stage training yields improvements beyond either stage alone, validating the effectiveness of the ViPER design.
>
> We also include additional representative cases in **Appendix D** to illustrate how ViPER enhances the model’s perceptual capabilities.

---

> ### Author Response · Authors · 2025-11-19
>
> ---
>
> **Weakness 3:**
> *The method relies heavily on self-generated data, which could potentially lead to issues with scalability and efficiency, especially for larger datasets. Future work could explore optimizations to reduce computational overhead and improve the model’s scalability without sacrificing performance.*
>
> **Response:**
> We agree with the reviewer that this is an insightful direction for future work. While ViPER achieves considerable performance gains with limited data and training cost through its carefully designed two-stage pipeline, we acknowledge the scalability limitations of the current framework.
>
> In future work, we plan to extend ViPER into an open-ended synthesis-training loop—moving beyond the fixed two-stage setup toward an infinite-phase framework that enables simultaneous data and model evolution. This would improve both scalability and efficiency while maintaining performance.
>
> ---
>
> **Weakness 4:**
> *The study focuses heavily on visual perception tasks, but the integration of visual understanding with textual reasoning could be further explored. The interplay between these two components could be better examined to understand their combined impact on multimodal reasoning tasks.*
>
> **Response:**
> We appreciate this thoughtful comment. Although our work emphasizes visual perception, we do not treat perception and reasoning in isolation. On the contrary, we believe that robust visual perception serves as the foundation for stronger reasoning capabilities.
>
> Several observations support this view:
> - As shown in Table 2, even without explicit reasoning-oriented training, Qwen-Viper achieves noticeable improvements on four logical and knowledge reasoning tasks, suggesting that enhanced perception can benefit reasoning.
> - In Section 4.3.2, we find that the first training stage facilitates the emergence of *"image thinking,"* indicating that textual reasoning chains also aid perceptual learning.
>
> We have also conducted additional experiments: during the RL stage, we attempted to incorporate **editing-based reasoning**—challenges that require logical inference based on perceived visual differences. For example:
>
> > **Target Object**: A small red chili bit stuck to the noodles near the chopsticks
> > **Question**: Based on what's being consumed, which flavor profile is confirmed to be present in this dish?
> > A. Citrus tang
> > B. Bitter herbs
> > C. Sweet and sour
> > D. Spicy heat
> > **Answer**: D. Spicy heat
> > **Analysis**: To solve this puzzle, one must carefully observe the noodles being held by the chopsticks in both images. In the second image, a small red chili piece has become attached to the noodles as they're being lifted from the bowl. This wasn't present in the first image. The presence of chili pieces in the dish confirms that it contains spicy heat as part of its flavor profile. The other flavor profiles listed (sweet and sour, bitter herbs, citrus tang) have no visible evidence in what's being picked up by the chopsticks. The red chili bit is a clear indicator of spiciness in the dish.
>
> However, we found that such reasoning-focused training was less effective than direct visual-operation prediction tasks. Interestingly, models trained only on visual-operation prediction still performed better on these reasoning questions. This suggests that perception is often the bottleneck in challenging visual tasks, and enhancing perception can enable more accurate, vision-grounded reasoning.
>
> In summary, although ViPER targets perceptual improvement, its benefits extend to multimodal understanding and reasoning, reinforcing the interdependence of perception and reasoning in VLMs.
>
>
> ---
>
> Thank you again for the thorough and constructive feedback! We hope our responses have adequately addressed your concerns. We would be deeply grateful if you could consider raising your score based on these clarifications. We look forward to further discussion if additional questions arise.

---

> ### Author Response · Authors · 2025-11-26
> **[Less Than A Week Remaining] Could you confirm whether our responses have addressed your concern?**
>
> Dear Reviewer YstH,
>
> We hope the above clarifications and the additional experiments sufficiently addressed your concerns. If you are satisfied, we kindly request you to consider updating the score to reflect the newly added results and discussion. We remain committed to addressing any remaining points you may have during the discussion phase.

---

> > ### Comment · Reviewer_YstH · 2025-11-28
> > **Response to rebuttal**
> >
> > I would like to thank the authors for their effort in preparing the rebuttal. These responses have largely resolved my concerns, and I have adjusted my score to be positive.

---

> > > ### Comment · Reviewer_YstH · 2025-11-28
> > >
> > > It seems that due to OpenReview’s restrictions, I am temporarily unable to edit my review. I will update the score once editing becomes available.

---

> ### Author Response · Authors · 2025-11-28
> **Thanks for the Positive Response**
>
> We are delighted to learn that our responses have resolved all the concerns. We sincerely appreciate your acknowledgment of our work and the decision to raise the score. Once again, we are grateful for your time and effort invested in reviewing our manuscript.

---

### Official Review · Reviewer_8UBN · 2025-11-05

**Soundness:** 3
**Presentation:** 3
**Contribution:** 3
**Rating:** 6
**Confidence:** 3

**Summary:**

This paper proposes ViPER, a self-bootstrapping training framework to improve fine-grained visual perception in VLMs without relying on external curated data. The core idea is a two-stage, coarse-to-fine framework: (1) Caption Self-Refining, in which a VLM captions an image, a diffusion model redraws the image from that caption, and the VLM critiques and then refines its caption by comparing the redraw with the original; (2) Visual-Operation Predicting, in which the model infers the edit applied to an image pair, thereby learning to attend to small, local changes. These yield a self-synthesized training set, Viper10K, and feed a coupled two-stage RL process, producing the Qwen-Viper variants from Qwen2.5-VL bases. On seven benchmarks encompassing single-image, multi-image, and hallucination tasks, Qwen-Viper shows ~1.6–1.7% average gains and up to 6.0% on fine-grained perception axes, with ablations supporting the need for both stages and for two-stage (vs. mixed) RL.

**Strengths:**

- This work presents a coarse-to-fine framework that separates holistic image captioning from localized visual-operation prediction. This design cultivates both scene-level understanding and region-specific editing reasoning, forming an executable synthesis pipeline.
- The system forms a closed loop that uses mismatches between the initial captions and the redrawn images as supervision, enabling label-free improvement.
- Across seven benchmarks covering multi-image tasks and hallucination diagnostics, the models improve consistently with largest gains on fine-grained perception up to +6.0% on the 7B variant.
- Ablations confirm that the two-stage design surpasses mixed RL approaches, with both stages providing complementary benefits.

**Weaknesses:**

- Results are limited to Qwen2.5-VL (3B/7B) while framework generality across other architectures/sizes is not demonstrated.
- Data synthesis relies on Qwen-Image and OmniGen2 to reconstruct/edit images without  cross-generator report or real-edit robustness tests, leaving a risk of generator-specific artifacts/shortcuts.

**Questions:**

- How sensitive are your results to the choice of BGE-M3?
- Have you run a supervised-only baseline on the same data/budget to quantify the gain from RL?
- Can you report results on at least one non-Qwen VLM (e.g., LLaVA/InternVL)?

---

> ### Author Response · Authors · 2025-11-19
>
> Thank you very much for your thoughtful evaluation and valuable feedback on our work. We would like to respond point by point to the issues you raised, in order to better demonstrate Viper’s contribution and effectiveness in enhancing the perceptual capabilities of vision-language models.
>
> ------------------
> **How sensitive are your results to the choice of BGE-M3?**
>
> Our experimental results are not dependent on the BGE-M3 model. As a text embedding model, it only serves to compute semantic similarity. We chose BGE-M3 because the model’s responses may contain multilingual content, and BGE-M3 is capable of computing embeddings for multiple languages. Other multilingual embedding models, such as multilingual-e5-large, would yield nearly equivalent results.
>
> ----
> **Reliability on generative models and potential risks**
>
> This is a valuable question. In Viper, we employ double mechanisms to minimize the impact of data noise introduced by generative models. First, for the challenging task of fine-grained editing, we introduce VLM-as-Judge to filter out generated results that do not meet the requirements. Implementation details and related prompts are provided in Appendix A.4. Second, for generator-specific artifacts that cannot be completely avoided, we fully circumvent their impact through reward design in the reinforcement learning (RL) stage. Although the VLM may produce “hallucinatory” self-critical content due to generative model bias during data construction, during training we only focus on recall based on semantic similarity. That is, such minimal hallucinated “refinements” will never be matched by the model’s rollout content, and their contribution to the advantage is zero, thus having no effect on the model’s gradient updates. In this way, we effectively eliminate the influence of generative model noise on the VLM.
>
> Moreover, when selecting generative models, we experimented with various models, including flux-kontext-dev, gpt-image1, Jimeng, Qwen-Image, OmniGen2, and others. We chose Qwen-Image for the first stage because of its ability to accurately reconstruct images containing text, preserving textual information to the greatest extent while maintaining strong instruction-following capabilities in overall generation quality. The second stage of data generation places greater emphasis on maintaining image consistency, and OmniGen2 is able to preserve high consistency in image content outside the edited areas in most cases, without introducing stylistic changes or inconsistencies in the main content. In fact, OmniGen2 is not the most capable editing model; later-released models such as Qwen-Image-Edit and Seedream4.0, as well as some closed-source models, possess stronger editing capabilities.
>
> -----
> **Have you run a supervised-only baseline on the same data/budget to quantify the gain from RL?**
>
> Yes, we have SFT Qwen2.5-VL-7B using the Viper-10K dataset . The results are reported below:
>
> | Model               | MMStar | RWQA | MME-RW | BLINK | Mantis-Eval | HallBench | CRPE | Overall |
> |---------------------|--------|------|--------|-------|-------------|-----------|------|---------|
> | Qwen2.5-VL-7B       | 63.9   | 68.5 | 57.4   | 56.4  | 75.1        | 52.9      | 76.4 | 64.4    |
> | +Viper10K (SFT)     | 64.5   | 68.2 | 57.9   | 56.8  | 75.6        | 53.2      | 76.1 | 64.6    |
> | Qwen-Viper-7B | 66.2   | 71.4 | 59.0   | 57.6  | 75.6        | 54.4      | 77.6 | 66.0    |
>
> As shown, the supervised fine-tuning approach did not yield significant improvements. We analyze this as follows: First, for pure supervised fine-tuning, the 10K-scale dataset provides limited improvement, as the data volume restricts the extent of gradient adjustment. Second, since the data does not include explicit chain-of-thought reasoning, the SFT approach cannot enable the model to exhibit visual reasoning emergence. Additionally, because the ground truth data contains minor noise from generator-specific artifacts, SFT causes the model to be influenced by this bias during training. Overall, these experimental results and analyses demonstrate that the success of the Viper framework relies not only on data design but, more importantly, on the close integration of data construction and downstream reinforcement learning, highlighting the superiority of the RL approach.
>
>
> -----
> **Can you report results on at least one non-Qwen VLM (e.g., LLaVA/InternVL)?**
>
> Thank you for this suggestion. Beyond the Qwen architecture, we are currently applying the Viper framework to VLMs from the InternVL series. The relevant training is underway, and we will update the experimental results once they are available.
>
>
> ------
> Finally, we sincerely appreciate your time in reviewing our response. If you feel that we have addressed your concerns, we would be very grateful if you would like to raise the score. If you have any further questions, we look forward to additional discussion and exchange.

---

> ### Author Response · Authors · 2025-11-29
>
> We have supplemented the experimental results of the ViPER framework on the InternVL series, with a particular focus on evaluating model performance on perception-intensive tasks. Specifically, we applied this method to both InternVL2.5 and InternVL3 models, with the following results:
> | Model               | MMStar | MMVP | RealWorldQA | Inst-IT (image) | HallBench | Overall |
> |---------------------|--------|------|-------------|-----------------|----------------|---------|
> | InternVL2.5-8B      | 62.8   | 68.0 | 70.1        | 69.7            | 50.1           | 64.1    |
> | InternVL2.5-Viper   | 65.1   | 71.7 | 71.6        | 72.5            | 52.8           | 66.7    |
> | InternVL3-8B        | 68.2   | 75.3 | 70.8        | 79.2            | 49.9           | 68.7    |
> | InternVL3-Viper     | 69.3   | 76.7 | 71.9        | 78.9            | 53.2           | 70.0    |
>
>
> From the experimental results, the ViPER framework achieves consistent improvements across both base model generations. Notably, while InternVL3 demonstrates significantly enhanced performance compared to InternVL2.5 due to more extensive training, ViPER still provides measurable gains. This confirms our method's effectiveness as an efficient incremental approach and reveals that even state-of-the-art reasoning models maintain potential for self-improvement.
>
> Furthermore, when combined with our previous results on the Qwen architecture:
> | Model               | MMStar | MMVP | RealWorldQA | Inst-IT(Image) | HallBench | Overall |
> |---------------------|--------|------|-------------|----------------|-----------|---------|
> | Qwen2.5-VL-3B       | 55.9   | 65.0 | 65.4        | 64.8           | 46.3      | 59.5    |
> | Qwen-Viper-3B       | 57.8   | 66.0 | 67.7        | 65.4           | 49.1      | 61.2    |
> | Qwen2.5-VL-7B       | 63.9   | 74.7 | 68.5        | 70.8           | 52.9      | 66.2    |
> | Qwen-Viper-7B       | 66.2   | 76.0 | 71.4        | 72.2           | 54.4      | 68.0    |
>
> we can confidently demonstrate that ViPER implements an architecture-agnostic self-evolution method. Notably, the performance gains are observed uniformly across different model sizes, further validating the effectiveness and scalability of our approach.
>
> Although our exploration of ViPER's underlying principles remains preliminary, the framework has already demonstrated considerable potential. By leveraging the core concepts of "self-evolution" and "promoting perception through generation", ViPER achieves substantial capability enhancements with minimal data requirements and low training costs. Future research may extend beyond fixed multi-stage tasks toward an infinite-phase framework where data and models co-evolve, enabling improved scalability and efficiency. Overall, ViPER demonstrates both promising effectiveness and profound research significance.

---

### Author Response · Authors · 2025-11-30
**Summary of Rebuttal**

Dear ACs and Reviewers:

We sincerely appreciate the constructive feedback provided by all reviewers and the additional efforts made by the ACs under these special circumstances.

----
First, we are delighted to note that the reviewers have provided **highly positive assessments and recognition**:

- The proposed self-evolutionary framework is innovative (Reviewer `YstH`), and the underlying idea and paradigm are interesting and novel (Reviewer `vNi5`).
- Our work offers an immediately practical and conceptually new route toward continual self-improvement of visual perception abilities (Reviewer `56QP`).
- Through robust experimental design (Reviewer `YstH`) and extensive ablation studies (Reviewer `8UBN`, Reviewer `vNi5`), we effectively demonstrate ViPER's performance improvements (Reviewer `YstH`).
- Our paper presents detailed illustrations of the method and experiments in a clear and well-structured manner (Reviewer `YstH`).

----

Based on this, the reviewers raised several insightful and constructive concerns and suggestions. We have conducted additional experiments and targeted analyses, which can be categorized into four main areas:

**Regarding the Idea and Methodology:**

- We have addressed Reviewer `56QP`'s concern about computational requirements from a **Pareto-optimality** perspective.
- Building on Reviewer `YstH`'s advice for future work, we analyzed a feasible and promising research direction for the ViPER methodology in terms of **scalability and efficiency**.

**Regarding Implementation:**

- In response to the shared concern of Reviewer `8UBN`, Reviewer `vNi5`, and Reviewer `56QP` regarding the reliability of the generative models, we provided a detailed explanation from the perspectives of data quality filtering and reward design. We elaborated on how our approach mitigates potential failures of the generative model through a dual mechanism of **VLM-as-Judge** and **zero-gradient contribution**.

**Regarding Experiments:**

- To address Reviewer `8UBN`'s concern, we supplemented the results of **supervised fine-tuning** on the Viper10K dataset and compared them with our two-stage RL results, demonstrating the superiority of RL in this paradigm.
- Following Reviewer `YstH`'s suggestion, we added **VL-Rethinker** as an additional baseline，a representative work also based on Qwen2.5-VL with reasoning enhancement, which further validates ViPER’s effectiveness in enhancing perceptual capabilities.
- Following Reviewer `vNi5`'s suggestion, we included **MMVP** and **Inst-IT** as additional benchmarks, enriching our experimental results. These benchmarks, which focus on fine-grained, instance-related perception capabilities, provide further evidence of our method’s effectiveness.
- Beyond the Qwen series models of varying scales, we supplemented the experimental results of applying the ViPER framework to the **InternVL** series, confirming that our method is **model-agnostic** and exhibits strong generalization. ViPER demonstrates **universal applicability** across VLMs with different architectures and performance, which adressed Reviewer `8UBN`'s concern.

**Regarding Writing:**

- Following Reviewer `vNi5`'s suggestion, we included the process of obtaining refined captions in **Appendix A.3**, further clarifying the complete implementation pipeline.

In summary, we have provided comprehensive and detailed responses to all the reviewers’ concerns, further improving the presentation and substantiating the outstanding contributions of our work. In the rebuttal version of the PDF, we have specifically supplemented the experimental results and key discussions, with the newly added content **highlighted in blue**.

----

It is worth noting that due to the early end of the discussion period, only Reviewer `YstH` provided a follow-up response. It is a pity that we had no chance to engage in further discussion with the other reviewers due to the accident. However, the only reply highly acknowledged the supplementary arguments we made in the rebuttal and indicated that we had **largely resolved the reviewer's concerns, and he would adjust the score to be positive**.

----

Once again, we extend our heartfelt thanks to all reviewers and the ACs for their diligent efforts and significant contributions.

Best regards,

The Authors

---

### Meta-Review · Area_Chair_Sd2P · 2025-12-17

**Summary:**

This paper studies the bottleneck of fine-grained visual perception in vision-language models and proposes ViPER, which is a self-bootstrapping framework built on a novel coarse-to-fine two-stage perception task. In more detail, ViPER integrates image- and instance-level reconstruction with a staged reinforcement learning strategy, forming a closed-loop training process driven by self-critiquing and self-prediction. Experiments confirm the effectiveness of the proposed method across diverse benchmarks.

The paper received review comments from four reviewers. They acknowledge that this work offers meaningful insights from the perspectives of the idea, method design, and empirical evaluation. Several concerns, mainly including the method description and the supplemented experimental results, are raised. The AC checked the paper, rebuttal, and reviewer feedback. The concerns are addressed properly. Therefore, the AC recommends acceptance. The authors are encouraged to incorporate the reviewers’ constructive feedback in the final version to strengthen the clarity and impact of this work.

**Reviewer Concerns:**

The concerns of the peer reviews were similar in many ways. Reviewers raised questions about the experiments, method issues, and unconvincing/unclear descriptions. A detailed rebuttal was provided subsequently. The AC checks the paper, questions, and answers, and thinks that the mentioned concerns have been addressed by the rebuttal.

**Reviewer Scores:**

- **Reviewer 8UBN.** The concerns include (1) results are limited to Qwen2.5-VL (3B/7B); (2) data synthesis relies on Qwen-Image and OmniGen2; (3) the sensitivity to the choice of BGE-M3; (4) results about the supervised-only baseline. The rebuttal includes detailed explanations and results to the concerns. Therefore, if the reviewer had been able to participate fully in the discussion, the score would be positive.

- **Reviewer YstH.** The concerns include (1) more recently released benchmarks; (2) more detailed ablations; (3) the concern of self-generated data; (4) the integration of visual understanding with textual reasoning. The rebuttal includes detailed explanations and results to the concerns. Therefore, if the reviewer had been able to participate fully in the discussion, the score would be positive. This actually was acknowledged by the reviewer.

- **Reviewer vNi5.** The concerns include (1) the reliance on a diffusion model; (2) an unclear method pipeline; (3) unclear experimental details, and more experimental results. Both results and explanations are provided. Therefore, the reviewer would be positive about the submission.

- **Reviewer 56QP.** The reviewer was concerned about the self-reinforcing loop of model-centric biases and computational costs of the method. The rebuttal provided explanations about the bias and claimed that the proposed method can approach the Pareto-optimal design. The feedback is overall convincing. Therefore, the reviewer would be positive about the submission.

---

### Decision · Program_Chairs · 2026-01-26

Accept (Poster)